# Development of a T Cell-Based COVID-19 Vaccine Using a Live Attenuated Influenza Vaccine Viral Vector

**DOI:** 10.3390/vaccines10071142

**Published:** 2022-07-18

**Authors:** Irina Isakova-Sivak, Ekaterina Stepanova, Victoria Matyushenko, Sergei Niskanen, Daria Mezhenskaya, Ekaterina Bazhenova, Elena Krutikova, Tatiana Kotomina, Polina Prokopenko, Bogdan Neterebskii, Aleksandr Doronin, Elena Vinogradova, Kirill Yakovlev, Konstantin Sivak, Larisa Rudenko

**Affiliations:** 1Institute of Experimental Medicine, 197022 Saint Petersburg, Russia; stepanova.ea@iemspb.ru (E.S.); matyshenko@iemspb.ru (V.M.); dasmez@iemspb.ru (D.M.); bazhenova.ea@iemspb.ru (E.B.); krutikova.ev@iemspb.ru (E.K.); kotomina@iemspb.ru (T.K.); prokopenko.pi@iemspb.ru (P.P.); rudenko.lg@iemspb.ru (L.R.); 2Joint-Stock Company «BIOCAD» (JSC «BIOCAD») Saint Petersburg, The Settlement of Strelna, 38 Svyazi Str., 198515 Saint Petersburg, Russia; niskanensa@biocad.ru (S.N.); bog2dan1@gmail.com (B.N.); doronin@biocad.ru (A.D.); vinogradovaev@biocad.ru (E.V.); 3Smorodintsev Research Institute of Influenza, 197376 Saint Petersburg, Russia; kirikus-fly@yandex.ru (K.Y.); konstantin.sivak@influenza.spb.ru (K.S.)

**Keywords:** SARS-CoV-2, COVID-19 vaccine, influenza virus vector, live attenuated influenza vaccine, T-cell epitopes, HLA-A2.1 transgenic mice, Syrian hamsters, cross-protection

## Abstract

The COVID-19 pandemic emerged in 2020 and has caused an unprecedented burden to all countries in the world. SARS-CoV-2 continues to circulate and antigenically evolve, enabling multiple reinfections. To address the issue of the virus antigenic variability, T cell-based vaccines are being developed, which are directed to more conserved viral epitopes. We used live attenuated influenza vaccine (LAIV) virus vector to generate recombinant influenza viruses expressing various T-cell epitopes of SARS-CoV-2 from either neuraminidase (NA) or non-structural (NS1) genes, via the P2A self-cleavage site. Intranasal immunization of human leukocyte antigen-A*0201 (HLA-A2.1) transgenic mice with these recombinant viruses did not result in significant SARS-CoV-2-specific T-cell responses, due to the immunodominance of NP_366_ influenza T-cell epitope. However, side-by-side stimulation of peripheral blood mononuclear cells (PBMCs) of COVID-19 convalescents with recombinant viruses and LAIV vector demonstrated activation of memory T cells in samples stimulated with LAIV/SARS-CoV-2, but not LAIV alone. Hamsters immunized with a selected LAIV/SARS-CoV-2 prototype were protected against challenge with influenza virus and a high dose of SARS-CoV-2 of Wuhan and Delta lineages, which was confirmed by reduced weight loss, milder clinical symptoms and less pronounced histopathological signs of SARS-CoV-2 infection in the lungs, compared to LAIV- and mock-immunized animals. Overall, LAIV is a promising platform for the development of a bivalent vaccine against influenza and SARS-CoV-2.

## 1. Introduction

Novel severe acute respiratory syndrome coronavirus (SARS-CoV-2) emerged in China in December 2019 [1,2] and by May 2022 has caused over half a billion cases of acute respiratory disease with over six million fatalities globally [3]. High transmissibility of the virus and the emergence of new SARS-CoV-2 antigenic variants have led to a tremendous global public health problem. To mitigate the coronavirus disease 2019 (COVID-19) pandemic, multiple vaccine candidates based on traditional and completely new technologies have been developed and licensed at an unprecedented speed (reviewed in [4,5]). A recent meta-analysis of the performance of COVID-19 vaccines in real-world settings demonstrated their high effectiveness against SARS-CoV-2-related diseases [6]. The vast majority of licensed vaccines, as well as vaccines under development, utilize Spike protein as a target antigen, since antibody directed to the receptor-binding domain (RBD) of Spike protein mediate virus-neutralizing activity, which strongly correlate with protection against symptomatic SARS-CoV-2 infection [7,8]. However, the main limitation of this antigen is its high variability among different lineages of SARS-CoV-2, which results in decreased effectiveness of the vaccines against emerging coronavirus variants [9].

Virus-specific T cells are another important mode of immunity which have the potential to limit disease severity and facilitate recovery [10]. The induction of memory T cells may solve the problem of waning antibody levels and the antibody escape by the new variants of concern, since these T cells are generally directed at conserved viral epitopes [10,11]. Although the first generation of COVID-19 vaccines based on adenoviral vectors or mRNA technologies induced substantial levels of cross-reactive T cells to spike epitopes [12,13,14], other viral targets such as Nucleocapsid or Membrane protein should be considered for vaccine development because these internal viral proteins contain important immunodominant T-cell epitopes with the potential of inducing long-lasting memory responses [14,15,16].

Thus far, several COVID-19 candidates have been designed specifically to induce broadly reactive and durable SARS-CoV-2 T-cell immunity. Some are composed of multiple T-cell epitopes derived from various structural and/or non-structural proteins [17,18], while the other approaches utilize whole viral proteins that are delivered to target cells either by DNA vaccination [19] or by viral vectors [20,21]. While epitope-based vaccines require potent carriers to induce strong T-cell responses, such as Toll-like receptor 1/2 agonist emulsified in Montanide [17], the delivery of SARS-CoV-2 proteins enriched with T-cell epitopes as a DNA vaccine or by the viral vectors enables intracellular processing and correct presentation of target epitopes to the immune system, thus forming relevant T-cell immunity. Earlier, we demonstrated the feasibility of using live attenuated influenza vaccine (LAIV) viruses as viral vectors for designing T cell-based vaccines against different human respiratory viruses, such as adenoviruses [22] and respiratory syncytial virus (RSV) [23,24]. Intranasal immunization of animals with these recombinant LAIV viruses led to effective formation of memory T cells specific to the target epitopes, and immunized animals were protected against corresponding infections in challenge experiments. Furthermore, the delivery of an immunodominant cytotoxic T lymphocyte (CTL) epitope of RSV directly to the respiratory tract using the LAIV virus vector not only induced robust RSV-specific lung-resident memory (T_RM_) CTL responses, but also augmented the influenza-specific T_RM_ responses [25].

In this study, we generated a panel of recombinant LAIV viruses encoding various SARS-CoV-2 antigenic fragments enriched with conserved T-cell epitopes and evaluated their replicative and immunogenic properties in vitro and in animal models. The most promising candidate was also assessed in Syrian hamsters for its potential to protect immunized animals against both SARS-CoV-2 and influenza virus.

## 2. Materials and Methods

### 2.1. Cells, Viruses and Peptides

#### 2.1.1. Cells

Madin–Darby canine kidney (MDCK) cells (ATCC CCL-34), African green monkey kidney (Vero) (ATCC CCL-81) and Vero E6 (ATCC C1008) cell lines were all cultured in Dulbecco’s Modified Eagle Medium (DMEM) supplemented with 10% fetal bovine serum (FBS) and 1× antibiotic-antimycotic solution (Gibco, Big Cabin, OK, USA) at 37 °C, 5% CO_2_. Vero-WHO (CB884) cells were cultured in Opti-PRO serum-free medium supplemented with 1× GlutaMax and 1× antibiotic-antimycotic (all from Gibco, Big Cabin, OK, USA).

#### 2.1.2. Viruses

A reassortant H7N9 live attenuated influenza vaccine virus was used as a viral vector to generate recombinant influenza viruses expressing T-cell epitopes of SARS-CoV-2 [26]. A homologous reassortant strain carrying hemagglutinin (HA) and neuraminidase (NA) from A/Shanghai/2/2013 (H7N9) and the remaining six genes from A/PR/8/34 (H1N1) virus (Sh/PR8) was obtained from the Center for Disease Control and Prevention (CDC, Atlanta, GA, USA). Influenza viruses were amplified in 10–11-day-old embryonated chicken eggs at 33 °C or 37 °C, clarified by low-speed centrifugation and stored at −70 °C in single-use aliquots. The infectious virus titer was assessed in eggs by end-point titration, calculated by the method of Reed and Muench and expressed as log_10_EID_50_/mL.

Two SARS-CoV-2 viruses isolated from COVID-19 patients in St. Petersburg, Russia, were provided by the Smorodintsev Research Institute of Influenza (St. Peterburg, Russia): hCoV-19/St_Petersburg-3524S/2020 (GISAID EPI_ISL_415710), from the original Wuhan lineage, and hCoV-19/Russia/SPE-RII-32759S/2021 (EPI_ISL_1789542), from Delta lineage. The SARS-CoV-2 viruses were propagated on Vero (ATCC CCL-81) cells cultured in DMEM supplemented with 2% FBS, 1× antibiotic-antimycotic solution (Gibco, Big Cabin, OK, USA) and 10 mM HEPES (DMEM/2%FBS) at 37 °C, 5% CO_2_. Cells were infected with the viruses at a multiplicity of infection (MOI) 0.005 or 0.01 for Wuhan and Delta strains, respectively. After 72-h incubation, cell supernatants were collected, spun down at 3500 rpm for 15 min and aliquoted in single-use vials. Sucrose gradient-purified viruses for in vitro stimulation of splenocytes were prepared as previously described [27].

SARS-CoV-2 infectious titers were determined by 50% Tissue Culture Infection Dose (TCID_50_) assay using 96-well cell-culture plates seeded with Vero-CCL81 cells. Ten-fold virus dilutions prepared on the DMEM/2%FBS were added to the wells and incubated at the 37 °C and 5% CO_2_ for 72 h. Virus-infected wells were determined visually by the presence of cytopathic effect (CPE). Infection titer was calculated by the Reed and Muench method and expressed in log_10_TCID_50_/mL. All procedures involving live SARS-CoV-2 were performed in the BSL-3 laboratory.

#### 2.1.3. Peptides

The influenza Len/17-specific peptide (NP_366–374_ ASNENMDTM) was chemically synthesized by Almabion Ltd. (Voronezh, Russia). The SARS-CoV-2-specific peptides (Appendix A) were chemically synthesized by IQ chemicals Inc. (Saint Petersburg, Russia). The purity of the obtained peptides was at least 95%, which was confirmed by analytical electrophoresis, analytical reverse phase high performance liquid chromatography and mass spectrometric analysis (MALDI). The CEF Control Peptide Pool was purchased from AnaSpec (Cambridge, UK). PepTivator SARS-CoV-2 Prot_N and PepTivator SARS-CoV-2 Prot_S were purchased from Miltenyi Biotec (Bergisch Gladbach, Germany). The peptides were reconstituted in dimethyl sulfoxide to a concentration of 1 mM and stored at −70 °C in single-use aliquots.

### 2.2. Designing of SARS-CoV-2 T-Cell Cassettes for Insertion into LAIV Genome

The analysis of existing data on human T-cell epitopes with confirmed processing deposited was performed using the Immune Epitope Database (IEDB, http://www.iedb.org/ (accessed on 25 February 2020)), as well as available literature data. The polyepitope cassettes were combined from S, N and/or M protein fragments enriched with selected T-cell epitopes. In order to optimize cellular processing of the cassettes, we carefully chose flanker regions of each epitope. The processing of resulting cassettes was assessed with IEDB tools to exclude undesirable neo-epitopes formation, and with allergen predicting tools (http://www.ddg-pharmfac.net/AllergenFP/ (accessed on 4 May 2020); http://www.allermatch.org/allermatch.py/form (accessed on 4 May 2020)) to exclude potential allergens.

### 2.3. Generation of Recombinant LAIV Viruses Expressing SARS-CoV-2 Polyepitope Cassettes

The cassettes were chemically synthesized by JSC BIOCAD (Saint Petersburg, Russia) and further cloned into NA or non-structural (NS) genes of H7N9 LAIV strain based on a master donor virus A/Leningrad/134/17/57 (H2N2) (Len/17), as previously described [22,23,24]. In brief, the cassettes were incorporated following the full-length N9 NA protein open reading frame (ORF), divided by a porcine teschovirus-1 (P2A) self-cleaving peptide [24,28]. In addition, the inserted cassette was followed by a duplicate fragment of NA gene, which is required for correct packaging. The insertion of the polyepitope cassettes into NS1 ORF, also separated by the P2A site, was accompanied by the truncation of the NS1 protein up to 126 residues. The polyepitope cassettes are supposed to be translated along with the NA or NS1 genes, followed by the cleavage of the two fragments via the P2A, which will facilitate independent intracellular processing of the inserted T-cell epitopes with proteasomes. The chimeric influenza virus genes were generated using Gibson cloning assembly method and inserted into pCIPolISapIT, a dual-promoter vector for reverse genetics of influenza virus. The XL-Gold E.coli cells were transformed with the resulting plasmids and then the plasmid DNAs were extracted using QIAGEN Plasmid Maxi Kit (QIAGEN, Hilden, Germany).

Recombinant influenza viruses carrying chimeric genes were rescued using the reverse genetics system developed for Len/17 master donor strain [29]. Briefly, 2 µg of each plasmid encoding HA and intact or chimeric NA gene of A/Anhui/1/2013 (H7N9) virus, an intact or chimeric NS gene and the remaining five genes of Len/17 virus were combined and transfected into Vero-WHO cells by electroporation using Neon Transfection System (Thermofisher Scientific, Waltham, MA, USA) with 100 µL Neon Kit according to manufacturer’s protocol. After electroporation, cells were incubated for 6 h at 37 °C, 5% CO_2_ in Opti-PRO serum-free medium supplemented with 1× GlutaMax (Gibco), followed by media change and the addition of 2.5 µg/mL trypsin (Sigma, Burlington, MA, USA). After 72 h incubation at 33 °C, 5% CO_2_, the cells were resuspended in culture medium and inoculated into 10–11-day old developing chicken embryos. After 72 h of incubation at 33 °C, the virus in chorioallantoic fluid was detected by hemagglutination test with 0.5% chicken red blood cells. Then, the rescued viruses were cloned by limiting dilutions in eggs, and the resulting clones were fully sequenced via Sanger sequencing. The selected clones with confirmed sequences were amplified in eggs and working stocks were prepared and stored at −70 °C for further analyses. Infectious titers of recombinant influenza viruses were determined in eggs and MDCK by end-point titration at 33 °C for 3 days and expressed as log_10_EID_50_/mL or log_10_TCID_50_/mL, respectively. We also assessed genetic stability of the rescued recombinant influenza viruses by performing ten serial passages in eggs, followed by Sanger sequencing of the chimeric viral genes.

### 2.4. Assessment of T-Cell Activation by Recombinant LAIV Viruses Using PBMCs of COVID-19 Convalescents

We used in vitro stimulation of PMBC specimens of COVID-19 convalescents with the chimeric LAIV viruses and subsequent intracellular cytokine staining (ICS) as indirect evidence for correct presentation of inserted SARS-CoV-2 T-cell epitopes delivered by the recombinant influenza viruses. For this purpose, peripheral blood mononuclear cells (PBMCs) were stimulated side-by-side with recombinant viruses, as well as with the LAIV vector, and the increased levels of cytokine-producing cells after stimulation with chimeric viruses relative to the vector control were interpreted as a positive response. Peripheral blood samples were collected from a cohort of blood donors who participated in the study of the persistence of SARS-CoV-2-specific T-cell responses after COVID-19 [27]. The study was approved by the local ethical committee of the Institute of Experimental Medicine (protocol #No 2/20 dated 7 April 2020), and all participants signed an informed consent. Twenty-five peripheral blood specimens were collected from twenty COVID-19 patients with disease onset from March 2020 until August 2020 and time post symptoms onset ranging from 1 to 5 months (Appendix A). During this period, the Wuhan D614G SARS-CoV-2 variant dominated circulation in Russia, suggesting that study participants were infected with this strain [30]. None of the participants was vaccinated with LAIV for the preceding five years, although the history of natural influenza infections in prior years could not be ascertained. PBMCs were isolated by standard procedures using Lymphocyte Separation Medium (Corning, Corning, NY, USA) and resuspended in 2 mL of warm CR-0 medium (RPMI-1640 medium supplemented with 1× antibiotic/antimycotic, 10 mM HEPES (Gibco, Big Cabin, OK, USA), 50 µM β-mercaptoethanol, 20 U/mL roncoleukin). For ICS, 2 × 10^6^ cells diluted in 50 µL of CR-0 medium were placed in a well of U-bottom plate and were stimulated by purified viruses in a dose of 3 MOI for 1 h at 37 °C under 5% CO_2_ conditions. After 1 h, an FBS-containing medium was added to a final concentration of 10% FBS. The cells were incubated with virus for 16–18 h at 37 °C, 5% CO_2_. Non-stimulated cells were incubated in the same conditions without stimulation. To stop intracellular transport BD Golgi Plug (BD Biosciences, San Jose, CA, USA) was added to a final dilution 1:1000. For positive control, phorbol myristate acetate (PMA) and ionomycin mixture (PMA + ionomycin Cell Activation Cocktail, Biolegend, San Diego, CA, USA) was used according to the manufacturer’s instructions. The cells were incubated for an additional 5 h at 37 °C, 5% CO_2_ and then stained with Zombie Aqua Fixable Viability Kit (BioLegend, San Diego, CA, USA) and antibodies to surface antigens: CD4 (CD4-APC-Alexa Fluor*750, clone 13B8.2, Beckman Coulter, Brea, CA, USA), CD8 (CD8-PC5.5, clone B9.11, Beckman Coulter), CD3 (CD3-PC7, clone UCHT1, Beckman Coulter), CD45RA (CD45RA-ECD, clone 2H4, Beckman Coulter), CD197 (CCR7) (CD197-FITC clone 150503, BD Pharmingen, San Diego, CA, USA). After staining, cells were washed twice and permeabilized with fixation/permeabilization kit (BD Cytofix/Cytoperm) according to the manufacturer’s manual. Cytokine staining was performed with PE Mouse-anti-human IFNγ antibody (clone B27, BD Pharmingen). After staining the cells were washed twice, fixed with Cytolast buffer (BioLegend, San Diego, CA, USA) and counted using Navios Flow Cytometer (Beckman Coulter). Data were analyzed with FlowJo software.

### 2.5. Immunization of HLA-A2.1 Transgenic Mice with LAIV/SARS-CoV-2 Vaccine Candidates

Groups of 11 eight- to twelve-week-old female HLA-A2.1 RANDOM TRANSGENIC mice (Taconic, Rensselaer, NY, USA) were inoculated intranasally (i.n.) with 10^6^ EID_50_ of each recombinant influenza virus and LAIV virus vector, in a volume of 50 µL, under isoflurane anesthesia. On day 3 p.i. four mice from each group were euthanized by an overdose of isoflurane, and nasal turbinates and lungs were collected for the assessment of viral replication in respiratory tract. Tissue homogenates were prepared in 1 mL PBS using a small bead mill (TissueLyser LT, QIAGEN, Hilden, Germany), centrifuged at 3500 rpm for 15 min, and viral titers in supernatants were determined by titration in eggs as described above. On day 21, the remaining mice were i.n. inoculated with the second dose of the same virus, and all animals were sacrificed ten days after revaccination. Blood samples were collected to study influenza virus-specific antibody responses in hemagglutination inhibition (HAI) assay and enzyme-linked immunosorbent assay (ELISA). In addition, splenocytes were isolated to study influenza- and SARS-CoV-2-specific cellular responses by ELISpot and ICS assays as specified below.

### 2.6. Immunization and Viral Challenge of Golden Syrian Hamsters

Groups of sixteen ten- to fourteen-weeks-old female Golden Syrian hamsters (Stolbovaya animal breeding nursery laboratory, Moscow region, Russia) were i.n. inoculated with 5 × 10^6^ EID_50_ of a recombinant LAIV/SARS-CoV-2 virus, as well as LAIV control virus, in a volume of 100 µL, under isoflurane anesthesia. On day 4 p.i., four hamsters were euthanized by an overdose of isoflurane to determine vaccine virus titers in the respiratory tract, as described above. The second vaccination was performed on day 21 of the study and three weeks after the second dose blood samples were collected from six animals in each group via the retro-orbital sinus, to determine influenza virus-specific antibody responses in ELISA. On day 45 of the study, immunized animals from each study group were divided into three groups of four hamsters which were infected i.n. with 100 µL of one of three viruses: (i) 10^6^ EID_50_ of Sh/PR8 influenza virus; (ii) 10^5^ TCID_50_ of SARS-CoV-2 (Wuhan lineage); and (iii) 10^5^ TCID_50_ of SARS-CoV-2 (Delta lineage). Influenza virus-infected animals were euthanized four days after infection, and respiratory tissues (NT and lungs) were collected and weighed for the assessment of virus loads. Tissue homogenates were titrated on MDCK cells and viral titers were expressed in log_10_TCID_50_/gram tissue. SARS-CoV-2-infected hamsters were monitored for weight loss and clinical symptoms (appearance, coat condition: 0—normal, 1—lack of grooming; interaction with other animals: 0—normal; 1—reduced; food consumption: 0—normal, 1—reduced; behavior on open area: 0—active, 1—reduced; response to pick up: 0—normal, 1—reduced) until day 5 p.i., when they were sacrificed for assessment of virological, immunological and pathological endpoints of protection. The lungs were perfused with 10 mL of PBS through the right ventricle and macroscopically assessed for the presence of pathological changes. One lung lobe was used for histopathological analyses, while the remaining tissue was weighed and homogenized to determine viral loads by TCID_50_ assay in Vero-E6 cells. In addition, nasal turbinates were collected to determine SARS-CoV-2 viral titers in Vero-E6 cells. Hamster splenocytes isolated on day 5 post-SARS-CoV-2 challenge were used to assess cellular recall responses by IFNγ ELISpot assay, as described below.

### 2.7. Assessment of Virus-Specific Antibody and Cellular Immune Responses

#### 2.7.1. Virus-Specific Antibody Immune Responses

Anti-influenza antibody levels were assessed in serum samples of immunized animals using HAI assay and/or ELISA, as described elsewhere [23]. Briefly, for HAI assay, serum samples were treated with receptor destroying enzyme (RDE) (Senka, Japan) and two-fold dilutions were incubated with four hemagglutination units of the H7N9 LAIV virus for 1 h at room temperature, followed by the addition of an equal volume of 0.5% chicken red blood cells. HAI titer was determined as the last serum dilution with complete inhibition of hemagglutination.

For influenza-specific ELISA, 50 ng/well of sucrose-purified LAIV virus were adsorbed on high-binding 96-well plates (Thermo, Waltham, MA, USA) overnight at +4 °C. Then, the wells were blocked with 1% of bovine serum albumin (BSA) in PBS, washed with PBS-Tween 20 (PBST), and two-fold serum dilutions were added to the wells and incubated for 1 h at 37 °C. After washing with PBST, an anti-mouse or anti-hamster IgG antibody conjugated to horseradish peroxidase (Sigma, Burlington, MA, USA) were added and the plates were incubated for another 1 h, followed by color development with 1-Step Ultra TMB-ELISA Substrate Solution (Thermo Fisher Scientific, Waltham, MA, USA). The results were read on xMark Microplate spectrophotometer (BioRad, Hercules, CA, USA) at a wavelength of 450 nm. Antibody titer was calculated as the maximum serum dilution with an optical density at 450 nm (OD_450_) value exceeding the control values at least two times.

SARS-CoV-2-specific ELISA was performed with sucrose gradient purified whole viral antigen (Wuhan and Delta) as described above for influenza virus, except that the SARS-CoV-2-coated plates were further inactivated by adding 2% buffered formalin and incubating the plates overnight at 4 °C, prior to treatment with BSA.

#### 2.7.2. Assessment of Cellular Responses by ICS Assay

Splenic tissues were homogenized with a pestle, followed by filtering through a 70 µm cell strainer (Becton Dickinson, Franklin Lakes, NJ, USA). Red blood cells were lysed by an ammonium-chloride potassium lysing buffer (Thermo Fisher Scientific) and the single cell suspensions were maintained in CR-0 media. Purified murine splenocytes were placed in U-bottom 96-well plates, at a concentration 10^6^ cells per well, in a volume of 50 µL. A sucrose gradient-purified LAIV virus was diluted in a CR-0 medium and 50 µL were added to the cells to reach a multiplicity of infection of 3. After incubation for 1 h at 37 °C and 5% CO_2_, 50 μL of CR-30 was added to achieve 10% final concentration of FBS. Non-stimulated cells were maintained under CR-10 medium. After 16–18 h incubation at 37 °C, 5% CO_2_, 50 µL of GolgiPlug solution at a 1:250 dilution (Becton Dickinson, Franklin Lakes, NJ, USA). At the same time, other wells were stimulated with SARS-CoV-2 peptide mix or NP_366_ influenza peptide, each peptide was diluted to a concentration 1 µg per well. Positive control wells were stimulated with PMA-ionomycin mixture. The plates were incubated for another five hours, followed by staining with a panel of fluorescently labeled antibodies: ZombieAqua, CD4-PerCP/Cy5.5, CD8-APC/Cy7, CD62L-BV421, and CD44-PE (all from BioLegend, San Diego, CA, USA). After fixation and permeabilization, the cells were stained for intracellular cytokines with antibodies IFNγ-FITC, TNFα-APC and IL-2-PE/Cy7 (all from BioLegend, San Diego, CA, USA). After staining, the cells were washed, fixed with Cytolast buffer and counted using Navios Flow Cytometer. The data were analyzed with FlowJo software.

#### 2.7.3. Assessment of Cellular Responses by ELISpot Assay

Splenocytes isolated from mice and Syrian hamsters were processed identically, except that different ELISpot kits were used to detect IFNγ production: Mouse IFN-γ Single-Color ELISPOT (Cellular Technology Ltd., Shaker Heights, OH, USA) and Hamster IFN-γ ELISpot Plus kit (Mabtech, Nacka Strand, Sweden) using the instructions of the manufacturers. Briefly, pre-coated with anti-IFNγ antibody ELISpot plates were washed 4 times with sterile PBS and then incubated with CR-10 media for 30 min at room temperature. A separate sterile 96-well plate was used to mix splenocytes with one of the desired stimuli: either purified influenza virus at an MOI 1.0, or purified SARS-CoV-2 virus at an MOI 0.1, or PepTivator S + N mixture (30 pmol per peptide), or specific peptides (1 µg per well). Then the media was removed from the ELISpot plate and the mixtures were added to each well, followed by 18 h incubation at 37 °C, 5% CO_2_. The detection of spots was performed according to the manufacturer’s protocol using detection antibody and ready-to use substrate solution. Color development was stopped by extensively washing in tap water and then the plates were left to dry overnight. Spots were counted in an AID vSpot Spectrum reader (Advanced Imaging Devices, Strassberg, Germany).

### 2.8. Histopathological Studies

Lung tissues were fixed for 48 h in a 10% neutral buffered formalin. For this study, lung tissues from a group of four naïve hamsters were used as a control. The whole lung longitudinal sections were made following RITA sampling guidelines [31]. Tissues were embedded in paraffin, thin-sectioned (3 µm) and stained with hematoxylin and eosin. Morphometric measurements and histological assessment included the mean airspace size and alveolar wall thickness measurement, as well as semiquantitative scoring of inflammatory lesions. The airspace size was evaluated using the mean linear intercept (MLI) of alveolar septa. MLI chord length was measured between intersections of the test line set with the alveolar surface, excluding septa and compressed meandering alveoli walls in 10 random non-coincident fields at ×200 magnification [32]. Alveolar wall thickness (AWT) was measured from leading edge to leading edge (l-l) in the same fields of view at ×200 magnification. The area taken by the inflammatory lesions was estimated roughly at ×50 magnification as a percentage of the whole lung section surface. All measurements were taken with ADF Image Capture ×64 and Leica DM1000 light microscope. Semiquantitative assessment was performed based on Carrol et al. with modifications [33]. Scoring criteria included:

(a)Airway pathology comprised of tree parameters: % airway affected (0—none, 1—<10%, 2—10–25%, 3—25–50%, 4—50+%); airway severity (0—minimal peribronchial/peribronchiolar mononuclear infiltrates, 1—mild peribronchitis/bronchiolitis, 2—mild to moderate mononuclear to mixed peribronchiolitis/lumens contain low numbers of inflammatory cells/multifocal single cell necrosis of airway epithelium, 3—moderate to marked mixed peribronchiolitis/large foci of bronchiolar epithelial necrosis/occasional atypical or multinucleated cells; 4—marked bronchiolitis and widespread epithelial necrosis +/− rupture of bronchiolar epithelium, and/or frequent atypical/syncytial cells); and bronchiolar epithelial hyperplasia (0—none, 1—sporadic bronchiolar epithelial hyperplasia < 10% section’s airways, 2—mild to moderate bronchiolar epithelial hyperplasia 10–25% section’s airways, 3 -widespread bronchiolar epithelial hyperplasia and/or multinucleated syncytial cells taking up 25+% section’s airways).(b)Lung/alveolar pathology comprised of tree parameters: % alveoli affected (0—none, 1—<10%, 2—10–25%, 3—25–50%, 4—50+%); alveolar severity (0—within normal margins (rare/minimal peribronchial/peribronchiolar mononuclear infiltrates), 1—mild peribronchiolar primary mononuclear inflammatory infiltrates, extending into adjacent alveolar septa/spaces, 2—mild to moderate, mononuclear to mixed inflammation (>3 cell layer), expands alveolar septa or spaces/obscures normal septal architecture, 3—moderate mixed interstitial inflammation, and/or alveolar damage characterized by type I pneumocyte necrosis/loss with replacement by hemorrhage, fibrin, edema, necrotic debris (reminiscent of hyaline membranes) and/or scattered atypical/syncytial cells, 4—marked alveolar inflammation (mixed), alveolar septal damage (all above) + loss of normal septal architecture with frequent syncytial cells); and type II pneumocyte hyperplasia (0—none, 1—scattered type II pneumocyte hyperplasia taking up <10% of the section, 2—mild to moderate type II pneumocyte hyperplasia taking up 10–15% + atypical multinucleated cells, 3—widespread type II pneumocyte hyperplasia taking up 25+% of the section).(c)Vascular damage comprised of two parameters: % vessels affected (0—none, 1—<10%, 2—10–25%, 3—25–50%, 4—50+%) and vascular/perivascular lesions (0—none, 1—multifocal perivascular edema/mild mononuclear perivascular inflammation, 2—moderate mononuclear to mixed perivascular inflammation, edema or fibrin with leukocytes occasionally transmigrating the vessel wall/multifocal endotheliitis, 3—severe mixed perivascular infiltration, expanding/replacing vessel wall and/or marked frequent endotheliitis).

### 2.9. Statistical Analysis

Data were analyzed using Graph Pad Prism 6 software. Statistically significant differences between several study groups were determined by one-way or two-way ANOVA with Tukey’s multiple comparison test. Differences between the levels of IFNγ-secreting T cells between LAIV vector and LAIV/SARS-CoV-2 recombinant viruses were assessed by non-parametric analysis using the Wilcoxon T-test. Statistical significance of the difference between two groups (evaluation of viral titers in vitro and in respiratory tracts) was determined by the Mann–Whitney U-test. *p* values of <0.05 were considered significant.

## 3. Results

### 3.1. Selection of SARS-CoV-2 Fragments Enriched with Conserved T-Cell Epitopes and Generation of Recombinant LAIV/SARS-CoV-2 Viruses

Structural proteins of SARS-CoV-2 (S, N, M) have been reported to be the main targets for T-cell response, as well as proteins synthesized early in the viral lifecycle [10,34,35,36,37,38]. Non-structural proteins of SARS and SARS-2 have high rate of conservancy, however, there are several warnings against using conservative parts of replicase, proteinase or other proteins in vaccine development. In studies with murine coronaviruses some replicase epitopes induced autoimmune neurological disorders because of its homology with myelin proteolipid protein [39,40].

This study was initiated at the beginning of COVID-19 pandemic, and the initial selection of fragments was based on sequence conservation between SARS-CoV and SARS-CoV-2 and data about experimentally confirmed epitopes deposited in IEDB and reported in literature. We were focused on epitopes with confirmed processing and presentation, this allows us to develop a method of assessment of epitope processing in a “model” situation. The main restriction of T-cell assays is poor allele coverage: the majority of well-studied epitopes are HLA-A*02:01-restricted because of availability of corresponding animal models. For SARS, large-scale analysis of binding of SARS peptides to human MHC was performed by different laboratories which deposited these data directly in IEDB [41,42,43,44,45]. It is supposed that a successful strategy of a T cell-based vaccine design is to combine experimentally confirmed T-cell epitopes with peptides with established MHC binding, in order to cover the majority of the human population [46].

Since the capacity of an influenza virus genome for the insertion of foreign transgenes is rather limited, we designed several variants of inserts encoding polyepitope cassettes ranging from ~100 to ~170 residues and comprising of different fragments of S, N and M proteins which contain a number of conserved epitopes with confirmed processing and immunogenicity, as well as peptides that were reported to form complexes with different human MHC molecules (Figure 1). More detailed information on the experimentally confirmed epitopes included in each selected fragment is provided in Appendix A. The cassettes were synthesized de novo and inserted into NA or NS1 genes of H7N9 LAIV virus, as shown in Figure 2. The insertion of the P2A site upstream of the transgene was supposed to facilitate the independent intracellular processing of SARS-CoV-2 epitopes (Figure 2). In this study, eleven rescued LAIV/SARS-CoV-2 variants have been evaluated in detail, five of which had the insertion into NA gene, while the other six variants had NS gene modified (Table 1). All recombinant LAIV viruses were genetically stable after ten passages in eggs, with no undesired mutations detected within the chimeric influenza genes.

### 3.2. Replicative Properties of Recombinant LAIV Viruses In Vitro and In Vivo

All recombinant viruses, as well as the control H7N9 LAIV strain, were grown in eggs and their titers in this substrate served as the main indicative of the impact of the inserted foreign fragments on viral replicative properties. Strikingly, only one out of 11 chimeric viruses (FluCoVac-28) had very similar titer to the LAIV vector, while the other variants showed the titer reduction from 1.4 to 3.0 log_10_EID_50_ (Table 1). The greatest impact was noted when the SARS-CoV-2 cassettes were inserted into the NS gene of the influenza virus, whereas modification of NA gene had a less pronounced effect on viral growth in eggs. Importantly, there was no correlation between the size of the inserted foreign gene fragment and the activity of viral growth in eggs. Similar results were obtained via the titration of recombinant LAIV viruses on MDCK cells: all chimeric viruses with the modified NS gene were not able to be replicated in this substrate, most likely due to the effect of NS1 truncation and not the insertion of the transgenes themselves.

Groups of four HLA-A2.1 transgenic mice were inoculated i.n. with the chimeric viruses at a dose of 10^6^ EID_50_ to assess their replicative properties in the upper and lower respiratory tract. On day 3 post inoculation, no live virus or only some residual quantities could be detected in the mouse lungs, suggesting the attenuated phenotype of all variants (Table 1). All vaccine candidates with modified NA gene replicated in nasal turbinates of mice at the level of classical LAIV virus which indicated the absence of a negative effect of such LAIV genome modification. In contrast, all viruses with the modified NS1 gene lacked active viral replication in the upper respiratory tract (Table 1), which was in line with previously published findings on the reduced replication of influenza viruses expressing truncated NS1 proteins in a mouse model [47,48].

### 3.3. Immunogenicity of the Recombinant LAIV-SARS-CoV-2 Viruses in Transgenic Mice

Since the inserted SARS-CoV-2 cassettes contained a number of confirmed HLA-A2 restricted CTL epitopes, we used HLA-A2.1 transgenic mice to evaluate humoral and cell-mediated immunity upon intranasal immunization with the recombinant LAIV/SARS-CoV-2 viruses. The chimeric viruses were tested in two independent experiments: the first study assessed six vaccine prototypes carrying cassettes #1 to #6, while the second experiment involved five variants expressing cassettes #7 to #10. Groups of 7 mice received two doses of the vaccines, 10^6^ EID_50_ each, with an interval of 21 days. On day 10 post-second dose, mice were sacrificed, and serum samples were used for assessment of antibody immune responses, while spleens were collected to study cell-mediated immunity.

#### 3.3.1. Influenza-Specific Antibody Immune Responses

Serum samples were assessed by influenza virus-specific HAI and ELISA assays. Despite the lack of detectable virus replication in some vaccine prototypes with modified NS genes, all vaccine candidates induced a virus-specific antibody, though the magnitude of these responses varied significantly between groups (Figure 3). These data suggest that the particular sequence of the inserted SARS-CoV-2 cassette can have an influence on the immunogenicity of the influenza virus vector itself. For example, viruses FluCoVac-13 and FluCoVac-14 have an identical size of the cassettes, which were inserted into the NS1 gene, but the antibody responses of the FluCoVac-13 prototype were 5–6 times higher than that of FluCoVac-14 strain (Figure 3). Another important observation is that the site of cassette insertion had no impact on the immunogenicity of the LAIV virus, since FluCoVac-27 and FluCoVac-32 candidates induced comparable influenza-specific antibody responses despite the differences in the replicative activity of the viruses in the mouse upper respiratory tract (Table 1). From the other side, the insertion of the shorter cassette into the NS gene could lead to a lower immunogenicity than the longer insert, as was evidenced by the pair of viruses FluCoVac-29 and FluCoVac-31 (Figure 1 and Figure 3).

#### 3.3.2. Influenza- and SARS-CoV-2-Specific T-Cell Responses

Isolated mouse splenocytes were stimulated with a sucrose-purified LAIV virus to assess influenza-specific cellular responses. Both IFNγ ELISpot assay and flow cytometry analysis confirmed the induction of robust T-cell responses to influenza virus, with some variations in the magnitude of the response, which generally correlated with influenza-specific antibody responses. In particular, the least immunogenic candidates FluCoVac-14 and FluCoVac-29 demonstrated the lowest levels of activated IFNγ-secreting T cells in both assays (Figure 4).

Strikingly, stimulation of the murine splenocytes with a mixture of SARS-CoV-2 peptides which corresponded to the inserted HLA-A*02:01 CD8 T-cell epitopes did not result in significant IFNγ production (Figure 4). We confirmed the expression of the HLA-A2 molecules on the mouse splenic cells using the PE Anti-HLA A2 antibody [BB7.2] (Abcam) (Appendix A), as well as the development of significant response to CEF control peptides, which contain immunodominant influenza M1 (GILGFVFTL) and PA (FMYSDFHFI) HLA-A2-restricted CTL epitopes (Appendix A). These data might indicate that the HLA-A2-restricted epitopes could be correctly processed and presented in this animal model; however, the immunodominance of influenza-specific epitopes can interfere with the response to the epitopes of interest. Moreover, T-cell responses to the mouse H2-Db-restricted influenza epitopes could also affect the immunogenicity of the inserted HLA-A2-restricted SARS-CoV-2 epitopes. Indeed, the HLA-A2.1 random transgenic mice originate from C57BL/6J background, which is known to generate a dominant CTL response to NP_366_ epitope of influenza virus [49,50]. In fact, the stimulation of splenic cells of immunized mice with the NP_366_ peptide revealed robust epitope-specific responses in all test groups (Appendix A). This study highlighted the importance of the development of relevant animal models for the assessment of human epitope-specific T-cell responses to the epitope-based vaccine prototypes against various diseases.

### 3.4. Assessment of Recombinant LAIV Viruses in ICS Assay with COVID-19 Convalescents Samples

We stimulated PMBCs isolated from COVID-19 recovered patients with the chimeric LAIV/SARS-CoV-2 viruses as a surrogate assay for the detection of correct processing and presentation of the inserted SARS-CoV-2 T-cell epitopes. Similar to mouse studies, the viruses were tested in two independent experiments with six and five variants carrying cassettes #1–6 and #7–10, respectively. A side-by-side stimulation of PBMCs with recombinant viruses and the LAIV vector revealed significantly higher levels of IFNγ-producing effector memory CD4 T cells in almost all tested vaccine candidates, relative to the LAIV control (Figure 5A). It should be noted that the levels of CD8 T_EM_ cells activated by the chimeric viruses significantly varied between blood donors (Figure 5B), with a trend of reducing the IFNγ-positive proportion with the increase of the time post symptom onset, which can be explained by a rapid decline of the SARS-CoV-2 specific CD8 T_EM_ within several months after recovery, whereas the CD4 T_EM_ subset can be maintained for a longer period [27]. The use of PBMCs of COVID-19 naïve individuals didn’t reveal significant differences in cytokine-producing CD4 and CD8 T cells between recombinant viruses and the LAIV control (Appendix A). Overall, the results of the ICS analysis suggest that the SARS-CoV-2 T-cell epitopes delivered by LAIV virus vector can be processed by the human immune system with a potential of inducing protective T-cell responses.

### 3.5. Immunogenicity and Protective Activity of a Selected LAIV/SARS-CoV-2 Vaccine Prototype in Syrian Hamsters

Based on in vitro and in vivo characterization of the developed recombinant LAIV/SARS-CoV-2 variants, we selected FluCoVac-28 candidate for its further characterization in a Syrian hamster model, because this virus replicated well both in eggs and MDCK cells (Table 1), induced influenza-specific immunity in mice at the level of classical LAIV variant (Figure 3 and Figure 4), and due to the potent activation of effector memory cytotoxic and helper T cells in in vitro tests on PBMCs of COVID-19 convalescents (Figure 5).

Groups of sixteen Golden Syrian hamsters were i.n. immunized with two doses of 5 × 10^6^ EID_50_ of the FluCoVac-28 and LAIV viruses, 3 weeks apart, while control animals received placebo (PBS). Both influenza viruses replicated to the high titers in the hamster nasal turbinates on day 3 p.i., while no replication was observed in the lung tissues, which confirmed the attenuated phenotype of the vaccine variants (Figure 6A). Serum samples collected from eight animals in each group on day 21 post second dose were assessed for the presence of influenza and SARS-CoV-2 whole virus-specific IgG antibody in ELISA, while four hamsters in each group were infected intranasally with 10^6^ EID_50_ of a homologous H7N9 PR8-based influenza virus, followed by tissue collection on day 4 post challenge for virus titration. As expected, both viruses induced robust influenza-specific immune responses (Figure 6B) and protected animals against challenge with a homologous influenza virus infection (Figure 6C).

Importantly, no SARS-CoV-2-specific serum IgG antibody responses were noted in immunized hamsters, since the T-cell cassette did not include immunodominant B-cell epitopes (Figure 7A). Immunized hamsters were also challenged with 10^5^ TCID_50_ of Wuhan and Delta variants of SARS-CoV-2 on day 21 after second immunization (four animals per challenge). Hamsters immunized with FluCoVac-28 prototype were protected against weight loss and clinical manifestation of the disease if compared to the PBS and LAIV groups after challenge with Wuhan variant (Figure 7B,C). Interestingly, body weight loss was reduced in both LAIV groups when the animals were infected with Delta virus (Figure 7B). Clinical symptoms of the Delta variant were not pronounced in the control PBS group, and no significant differences were observed by this parameter between the test vaccines (Figure 7C). Interestingly, significant reduction of viral titers in the FluCoVac-28 immunized hamsters was detected only in the nasal turbinates of animals challenged with both SARS-CoV-2 variants (Figure 7D). Although a three-fold reduction was noted in the mean viral pulmonary titers in this test group relative to the mock-immunized animals, these differences were not statistically significant, probably due to the low number of animals in each group (*n* = 4).

The protective effect of vaccination was further studied by histopathological assessment of SARS-CoV-2-induced lung pathology. After challenge with Delta variant, inflammatory lesions in the lungs of mock-immunized hamsters were located predominantly around large bronchi and vessels, characterized by extensive involvement of the lung parenchyma, pronounced bronchiolitis/bronchitis and endotheliitis (Figure 8A and Appendix A). The bronchiolar epithelium was partially necrotic and desquamated into the lumen of the large bronchi. Along with necrobiotic changes, foci of regenerative hyperplasia of the respiratory epithelium were determined. Peribronchial and perivasal lymphohistiocytic infiltrates spread over a vast area of the lung parenchyma, lysing the walls of the alveoli and respiratory bronchioles. On the periphery, thickening and edema of the interalveolar septa, hyperplasia of type II alveolocytes were determined. The lumen of the alveoli contained a mixture of cell debris, polymorphocellular leukocytes, necrotic alveolocytes, fibrin and erythrocytes (Figure 8A and Appendix A). Interestingly, lung pathology of the LAIV-immunized hamsters was less pronounced compared to the control animals. In particular, inflammatory infiltration was located focally around several large bronchioles, characterized by a moderate amount of involvement of the lung parenchyma. The spread of inflammatory infiltration was segmental, while bronchioles and large vessels were affected in the apical segments only. Minor peribronchial infiltrations and foci of thickening of the interalveolar septa with lymphocytic infiltration were determined, which were sporadic (Figure 8). Nevertheless, lungs of the hamsters immunized with the recombinant LAIV/SARS-CoV-2 virus were the least affected, as their histoarchitectonics was comparable to the intact animals. Single foci of bronchiolitis and inflammatory infiltration of a moderate volume, as well as an insignificant volume of interalveolar septa thickened due to lymphoplasmacytic infiltration, were found in some lung lobes (Figure 8A and Appendix A). A semi-quantitative analysis of lung pathology revealed significant reduction in scoring values of airway, lung/alveolar and vascular damage in the FluCoVac-28 group compared to the mock-immunized animals (Figure 8B). Interestingly, the airway and vascular scores were also reduced in the LAIV-immunized hamsters compared to the PBS group (Figure 8B), which correlated with the reduced weight loss (Figure 7B), suggesting some degree of cross-protectivity afforded by the classical LAIV virus against SARS-CoV-2 infection. Noteworthy, this cross-protection was not observed when the immunized hamsters were challenged with SARS-CoV-2 Wuhan variant: both mock- and LAIV-immunized hamsters had severe lung lesions on day 5 after challenge, in form of broncho-interstitial pneumonia characterized by diffuse alveolar damage (Appendix A). These differences can be explained by the varying degree of pathogenicity of different SARS-CoV-2 variants in this animal model [51].

To find possible mechanism of protective effect of the chimeric LAIV/SARS-CoV-2 virus against SARS-CoV-2, we measured the levels of virus-specific cell-mediated responses by stimulating hamster splenocytes collected five days post challenge with live SARS-CoV-2 or PepTivator, followed by counting IFNγ-secreting cells by ELISpot. Interestingly, in animals from the naïve group significant numbers of spots were detected in both challenge experiments (up to 100 and 200 spots per 500,000 cells for Delta and Wuhan challenge, respectively) (Figure 9), suggesting that the immune system has been activated by day 5 after infection. Nevertheless, the FluCoVac-28-immunized hamsters demonstrated approximately three-fold higher levels of SARS-CoV-2-specific cells, which indicated reactivation of memory T-cell responses established by primary vaccination (Figure 9). Of note, the LAIV-immunized hamsters also demonstrated increased IFNγ production after challenge with SARS-CoV-2, most probably due to the nonspecific activation of SARS-CoV-2-specific T cells by the presence of robust T-cell immunity to influenza after LAIV immunization.

## 4. Discussion

Despite the successful implementation of first generation COVID-19 vaccines to reduce the pandemic burden, intensive research continues in an attempt to design vaccine candidates with improved characteristics. Such improved vaccines should have broader reactivity for protection against emerging coronaviruses and, if possible, combine protection against several most important respiratory pathogens. Due to the proven ability of SARS-CoV-2 to evolve antigenically and escape neutralizing antibody immunity raised to the prior infection or vaccination with Spike-based vaccines, the development of vaccines targeting to the mutationally constrained T-cell epitopes is considered highly promising [52]. Furthermore, there is evidence that soon after infection with SARS-CoV-2, especially the mild form, people do not have detectable virus-specific antibodies, although T-cell immunity is quite robust, either because no antibodies have been produced or they were very short-lived, whereas T-cell memory lasts much longer [53]. Indeed, analysis of blood samples taken from survivors of the previous coronavirus outbreak caused by SARS-CoV found that the virus-specific memory T cells can persist as long as 11 years post-infection [54], suggesting that the T cell-based COVID-19 vaccines could induce durable protection.

Multiple studies involving PBMCs of COVID-19 convalescents and human blood samples collected prior to the emergence of SARS-CoV-2 identified immunodominant and cross-reactive T-cell epitopes [38,55,56,57,58,59,60,61]. In addition, in silico immunoinformatics analyses have identified promising T-cell epitopes that can bind to diverse HLA alleles and cover up to 100% global population [46], which also significantly contributed to the development of broadly protective T-cell based COVID-19 vaccines.

In terms of vaccine development, the most promising studies for T-cell targeted vaccine are T-cell studies, which confirm not only the ability of peptide to bind to MHC molecule, but also to induce T-cells response, cytotoxicity, provide protection and form a population of memory T cells. However, due to the limited accessibility of HLA-transgenic animals, the immunogenic HLA-A*02:01-restricted epitopes are presented in IEDB database in a disproportionately greater number than the epitopes restricted by other alleles, which are predominantly characterized by the ability to bind to the corresponding human MHCs, but not by the immunogenic properties [41,44,45]. In vaccine design, the best strategy seems to be to combine experimentally confirmed T-cell epitopes with peptides with established MHC-binding to cover all human population. Therefore, in the current study we designed several cassettes comprising of various conserved SARS-CoV-2 regions enriched with the T-cell epitopes, which were either assembled from short parts of S, M, and N proteins carrying immunogenic CTL epitopes, or contained more prolonged SARS-CoV-2 fragments that included both cytotoxic and helper T-cell epitopes. In this cassette design we kept in mind that the order of epitopes and specific flanking regions can significantly affect epitope processing [62,63]. Furthermore, possible presence of epitopes with allergic properties was closely monitored to avoid undesired immune reactions to vaccination.

A limitation of the study is that some immunodominant and conserved SARS-CoV-2 T-cell epitopes were not included into our T-cell cassettes as they were discovered after the cassette design was completed and further optimization on the composition of T-cell epitopes within the recombinant influenza viruses should be continued to update epitope content in the T-cell cassettes. Nevertheless, the main advantage of the T cell-based vaccines is that the target epitopes can be combined into relatively short multiepitope cassettes that can be delivered even by viral vectors with low genome capacity, such as influenza virus.

In this study, we chose LAIV as a delivery vehicle as it was previously shown to be effective inducer of T-cell immunity in humans [64,65] and because our previous studies demonstrated the possibility of constructing LAIV-based viral vectored vaccines that efficiently induced T-cell responses to the inserted epitopes in mice [22,23,25]. In particular, vaccine prototypes developed in this study were based on potentially pandemic H7N9 LAIV strain as a viral vector, which was proven to be safe and immunogenic in humans [66]. Additional advantage is that the vaccine can be more effective in humans due to the lack of pre-existing immunity. Nevertheless, development of such vaccines based on seasonal LAIV viruses is also of high priority as the vaccine can be used as a bivalent vaccine for combined prophylaxis of the two most important respiratory pathogens, if the SARS-CoV-2 acquire similar seasonality as influenza.

We probed two strategies to generate a panel of recombinant LAIV viruses, with modification either NA or NS gene of the influenza virus. Both strategies have been successful in induction robust T-cell immunity to the inserted immunodominant epitopes in BALB/c [23] and C57BL/6J mice [22]. The NA modification doesn’t impact the influenza protein itself, whereas NS1 protein was truncated to 126 residues in an attempt to increase vaccine’s immunogenicity, as was demonstrated with other influenza strains [67,68]. However, in our study there was no significant enhancement of influenza-specific immune responses in NS-modified recombinant viruses compared to NA-modified. Moreover, the lack of detectable viral replication of the NS-modified strains interfered with robust antibody and cellular immunity, suggesting that truncation of the NS1 protein of a cold-adapted influenza virus makes it overattenuated and leads to the reduced immunogenicity. Nevertheless, the most important observation from the mouse study is that the particular content of the inserted foreign cassette had the most prominent effect on the influenza-specific immune responses than the size of the insert and the influenza virus gene modified. Although we did not evaluate the anti-influenza protective effect of recombinant LAIV/SARS-CoV-2 in mice in the current study because of the limited number of available animals, our previous results assume that NA- and NS-modified recombinant LAIVs can induce sterile immunity to influenza [22,23].

The lack of adequate small animal models to assess immunogenicity of human T-cell epitopes makes the development of COVID-19 T cell-based vaccines very challenging, resembling the hurdles researchers have encountered in testing epitope-based universal influenza vaccines [69], as the available transgenic animals usually express only one MHC allele, which may not reflect the real diversity of the human population. Since our constructs included confirmed HLA-A*02:01 CD8 T-cell epitopes, we studied immunogenicity of the recombinant LAIV/SARS-CoV-2 viruses on HLA-A2.1 transgenic mice, however, no significant SARS-CoV-2-specific responses were observed after intranasal immunization, most probably due to the immunodominance of influenza-specific mouse T-cell epitopes. For example, a side-by-side screening of T-cell epitopes in HLA-A2.1 transgenic mice and HLA-A*02:01-positive subjects who received a vaccine against vaccinia virus found that only 46% epitopes were detected in both systems, and the responses in transgenic mice against epitopes originally identified in mice were much stronger than for epitopes originally detected in humans, and vice versa [70].

Since the processing of T-cell cassettes containing SARS-CoV-2 epitopes in the influenza virus was not indicative in the humanized mice, we used an indirect in vitro test using PBMCs of people who recently recovered from COVID-19 to confirm processing of the inserted SARS-CoV-2 epitopes. It should be noted that all study participants had a history of exposure to influenza antigen, and some had received inactivated vaccine during their lifetime, but none had been immunized with LAIV. In this study, we compared the response to the LAIV antigen alone (i.e., the viral vector) with the response to recombinant LAIV viruses containing SARS-CoV-2 inserts and considered the differences in response to the two stimulating agents as a response to the inserted SARS-CoV-2 epitopes. In our study, the LAIV vector was based on the H7N9 influenza virus, which does not circulate in humans, so memory T-cell responses to HA and NA epitopes in our participants were not expected to be significant. In addition, LAIV was based on the Leningrad/17 H2N2 virus, and H2N2 viruses have not circulated in humans since 1968. All of our participants were born after 1968, so T-cell responses to internal proteins were also expected to be low.

In vitro stimulation of PBMC samples with most of tested live recombinant LAIV/SARS-CoV-2 viruses resulted in significant activation of cytotoxic and helper memory T cells in the group of COVID-19 convalescents, but not in the naïve subjects. These data indicate that the recombinant influenza viruses indeed can successfully express the inserted epitopes in infected cells, which are then presented by MHC-I and MHC-II molecules. The diverse levels of activated memory T cells in PBMC samples of COVID-19 convalescents can be explained by the varying time post symptoms onset, as these levels decrease over time [27] and also probably by the differences in the HLA alleles in our study participants.

Based on in vitro and in vivo characterization, we selected a promising vaccine prototype for further evaluation in pre-clinical studies. This candidate, FluCoVac-28, had a high ability to grow in both eggs and MDCK cells, induced robust influenza-specific humoral and cellular responses and was able to activate memory T cells in PBMCs of COVID-19 recovered patients. Moreover, the inserted T-cassette contained rather extended fragments of coronavirus proteins enriched with T-cell epitopes of various HLA restriction, including such immunogenic epitopes as RLQSLQTYV [58,71,72], RLDKVEAEV [72,73], GMEVTPSGTWLTYTGAIKLD [36,74], and others.

This selected prototype was further studied in Golden Syrian hamsters. Replicative properties of the recombinant virus and its ability to induce influenza-specific immunity were identical to the unmodified LAIV virus. Furthermore, the FluCoVac-28 protected immunized hamsters against both influenza and SARS-CoV-2 infections, which was confirmed by reduced weight loss, milder clinical symptoms and less pronounced histopathological sighs of SARS-CoV-2 infection in the lungs, compared to LAIV- and mock-immunized animals. Interestingly, although the recombinant virus was not effective in significant reduction of viral loads in the lungs of challenged Syrian hamsters, the animals were protected against lung immunopathology. These data are in line with other studies which found that virus-neutralizing antibodies are more effective in virus control than the T cells, but CD4 and CD8 T cells can effectively protect hamsters against antibody-resistant SARS-CoV-2 variants, without inducing lung immunopathology [75].

Although the data on hamster T-cell epitopes of SARS-CoV-2 is still limited, the induction of SARS-CoV-2-specific cellular responses and the observed protective effect against two antigenically diverse coronaviruses suggest that the inserted prolonged fragments of S and N proteins include a class I- and/or class II-restricted T cell subpopulations specific for this animal model. Interestingly, we observed some degree of non-specific protection afforded by classical LAIV virus against SARS-CoV-2 Delta variant. This phenomenon can be explained by the existence of cross-reactive epitopes in influenza virus and SARS-CoV-2 which can bind to the same T-cell receptors even in the absence of 100% sequence homology [76]. Furthermore, there is evidence that pre-existing T-cell immunity to influenza in healthcare workers correlated with cellular responses to SARS-CoV-2 after COVID-19 [77], suggesting that influenza immunity can contribute to the responses to SARS-CoV-2 infection and vaccination. Interestingly, similar results of nonspecific protection were observed in a study of another LAIV-based SARS-CoV-2 vaccine in Syrian hamsters [78]. The different level of non-specific protection of LAIV against Wuhan and Delta virus observed in the hamster challenge experiments can be explained by different pathogenicity of the two SARS-CoV-2 variants for these animals, as we noted the milder course of the disease after infection with Delta strain, compared to the Wuhan virus, and the non-specific activation of T cells could be more effective in case of lighter infection.

## 5. Conclusions

Overall, in this study we designed a T cell-based COVID-19 vaccine by inserting a polyepitope SARS-CoV-2 cassette into genome of LAIV virus. Assessment of the selected LAIV/SARS-CoV-2 variant in a Syrian hamster model of influenza and SARS-CoV-2 infections demonstrated the protective effect against both viruses, suggesting its potential for application as a bivalent vaccine against SARS-CoV-2 and influenza. Whether this vaccine can protect against co-infection with the two viruses is the area of our further research. Nevertheless, in this pilot study, it was important to demonstrate the effect against each infection separately to ensure that the vaccine could be effective if the vaccinated individual were to contract the infections at different times, which is more likely than infection with two viruses simultaneously. Furthermore, we herein tested only variants carrying a single polyepitope cassette, but it is possible to create recombinant viruses carrying several cassettes in different genes of LAIV virus, which can significantly increase the protective potential of the vaccine against SARS-CoV-2. In particular, we are also developing bivalent vaccine candidates based on LAIV expressing the RBD Spike fragment to induce antibody-mediated protection, and the ultimate goal of developing an effective bivalent vaccine is to induce sustained antibody and T-cell responses to SARS-CoV-2 by delivering selected B and T cell cassettes into target cells using the LAIV viral vector.

## Figures and Tables

**Figure 1 vaccines-10-01142-f001:**
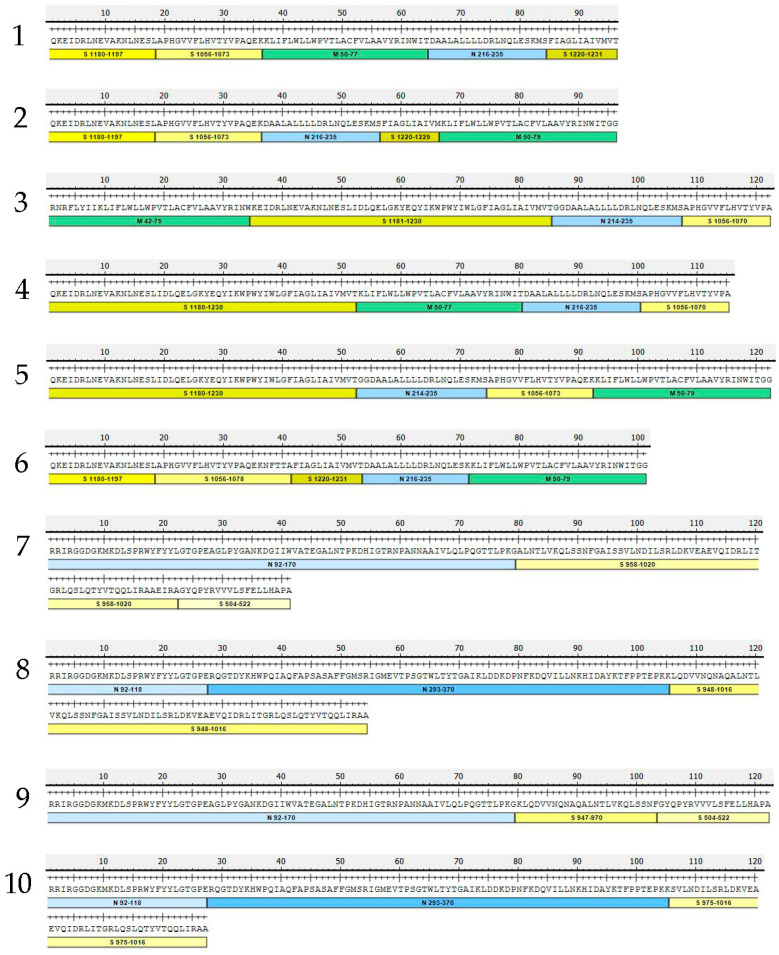
A schematic representation of selected cassettes comprising a panel of promising SARS-CoV-2 T-cell epitopes with confirmed processing and immunogenicity, as well as peptides that were reported to form complexes with different human MHC molecules. The cassettes were chemically synthesized and incorporated into the genome of LAIV virus.

**Figure 2 vaccines-10-01142-f002:**
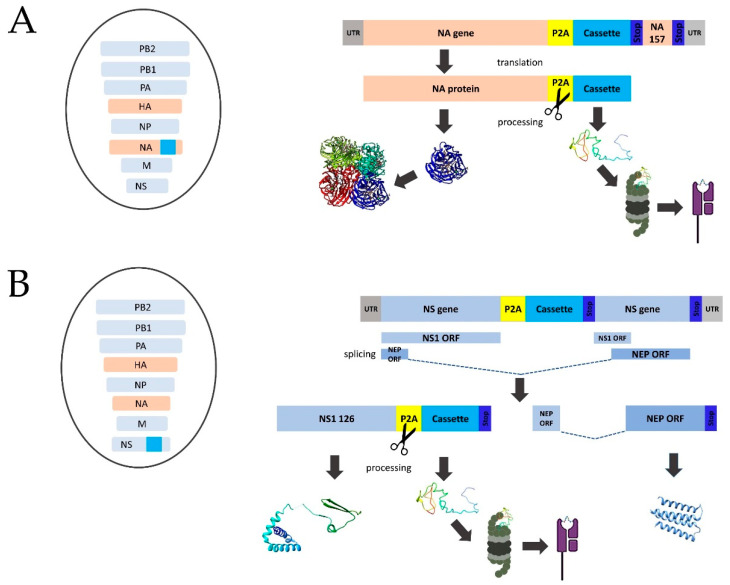
The generation of recombinant LAIV viruses expressing SARS-CoV-2 T-cell cassettes. (**A**) Schematic representation of modified NA gene of influenza virus. (**B**) Schematic representation of modified NS gene of influenza virus. The left panel represents the genome compositions of the rescued recombinant influenza viruses with Len/17 genes shown in light blue and H7N9 HA and NA genes shown in ochre. The SARS-CoV-2 polyepitope cassette inserts are shown in dark blue. P2A: a self-cleavage site (GSGATNFSLLKQAGDVEENPG↓P).

**Figure 3 vaccines-10-01142-f003:**
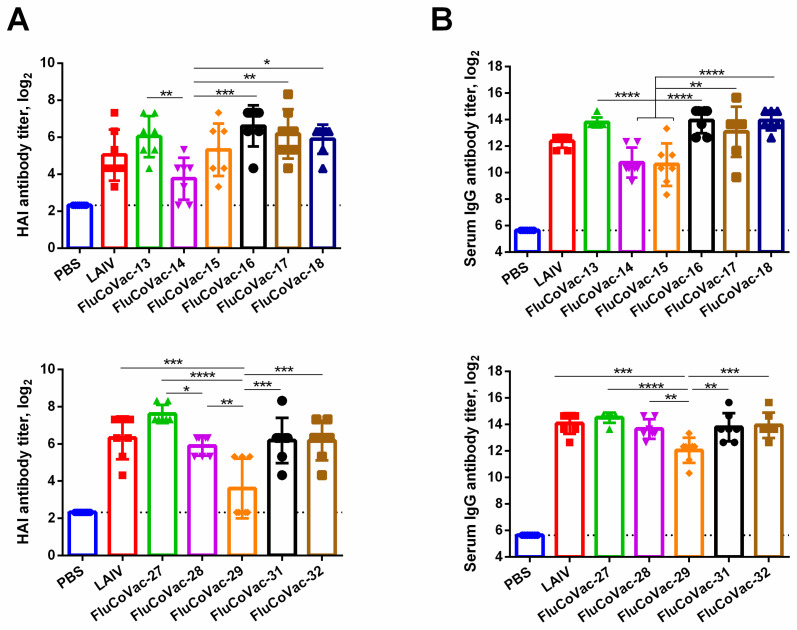
Antibody responses to the whole influenza virus in HLA-A2.1 transgenic mice immunized with studied recombinant LAIV viruses. Mice were i.n. inoculated twice with 10^6^ EID_50_ of each test virus within a 3-week interval. On day 10 post second dose serum samples (*n* = 7) were collected and influenza virus-specific antibody were assessed by (**A**) HAI assay and (**B**) ELISA to the whole influenza virus antigen. Upper and lower panels demonstrate the results of experiment #1 and #2, respectively. Data were analyzed by one-way ANOVA with Tukey’s post-hoc multiple analyses test. *—*p* < 0.05; **—*p* < 0.01; ***—*p* < 0.001; ****—*p* < 0.0001.

**Figure 4 vaccines-10-01142-f004:**
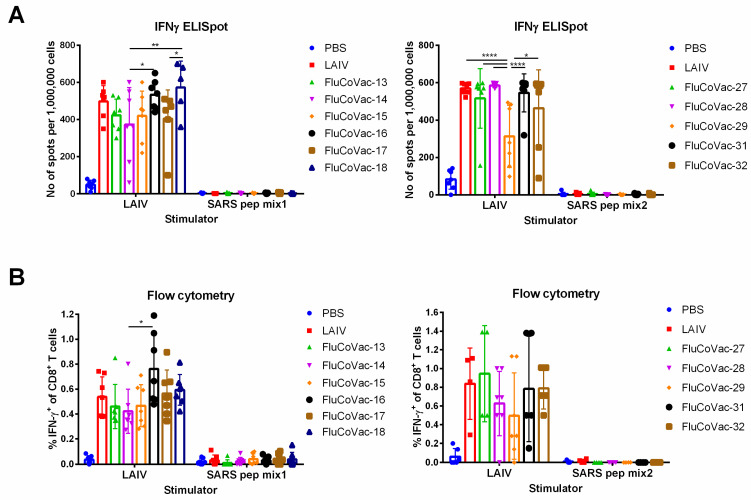
Virus-specific cellular immune responses in HLA-A2.1 mice. Transgenic mice were i.n. inoculated twice with 10^6^ EID_50_ of each test virus within a 3-week interval. On day 10 post second dose mouse splenocytes were collected and virus-specific cellular responses were assessed by (**A**) ELISpot analysis or (**B**) ICS after stimulation with whole influenza virus or a mixture of SARS-CoV-2 peptides. Data were analyzed by two-way ANOVA with Tukey’s post-hoc multiple analyses test. *—*p* < 0.05; **—*p* < 0.01; ****—*p* < 0.0001.

**Figure 5 vaccines-10-01142-f005:**
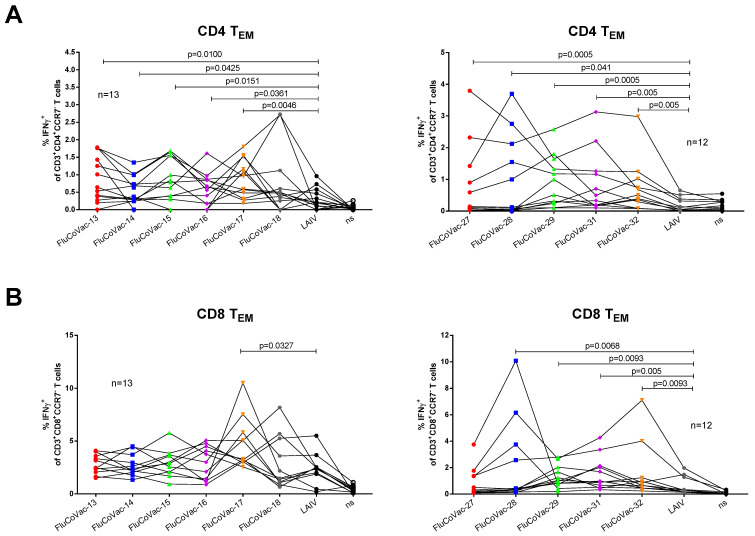
The levels of cytokine-producing effector memory T cell subsets in PBMC samples were stimulated with the studied recombinant LAIV/SARS-CoV-2 viruses. (**A**) CD4 T_EM_ cells. (**B**) CD8 T_EM_ cells. Upper and lower panels demonstrate the results of experiment #1 and #2, respectively. Differences between the levels of IFNγ-secreting T cells between LAIV vector and LAIV/SARS-CoV-2 recombinant viruses were assessed by non-parametric analysis using the Wilcoxon matched pair test.

**Figure 6 vaccines-10-01142-f006:**
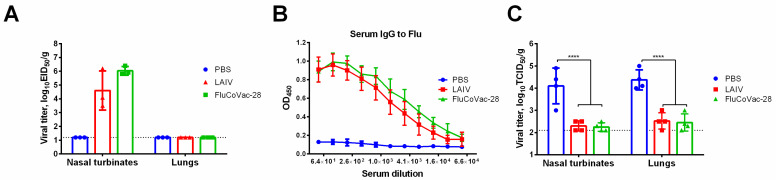
The replication and anti-influenza immune response, as well as protection of the studied viruses in Golden Syrian hamsters. Groups of Syrian hamsters were inoculated twice with 5 × 10^6^ EID_50_ of LAIV or FluCoVac-28 or were mock-immunized with PBS with 3-week interval. (**A**) Infectious viral titers in the respiratory tissues on day 3 after first inoculation. (**B**) Influenza-specific antibody responses as determined by ELISA against whole LAIV virus in serum samples collected three weeks after the second dose. (**C**) Infectious titers in the respiratory tissues on day 4 after challenge with homologous influenza virus. Data were analyzed by two-way ANOVA with Tukey’s post-hoc multiple analyses test. ****—*p* < 0.0001.

**Figure 7 vaccines-10-01142-f007:**
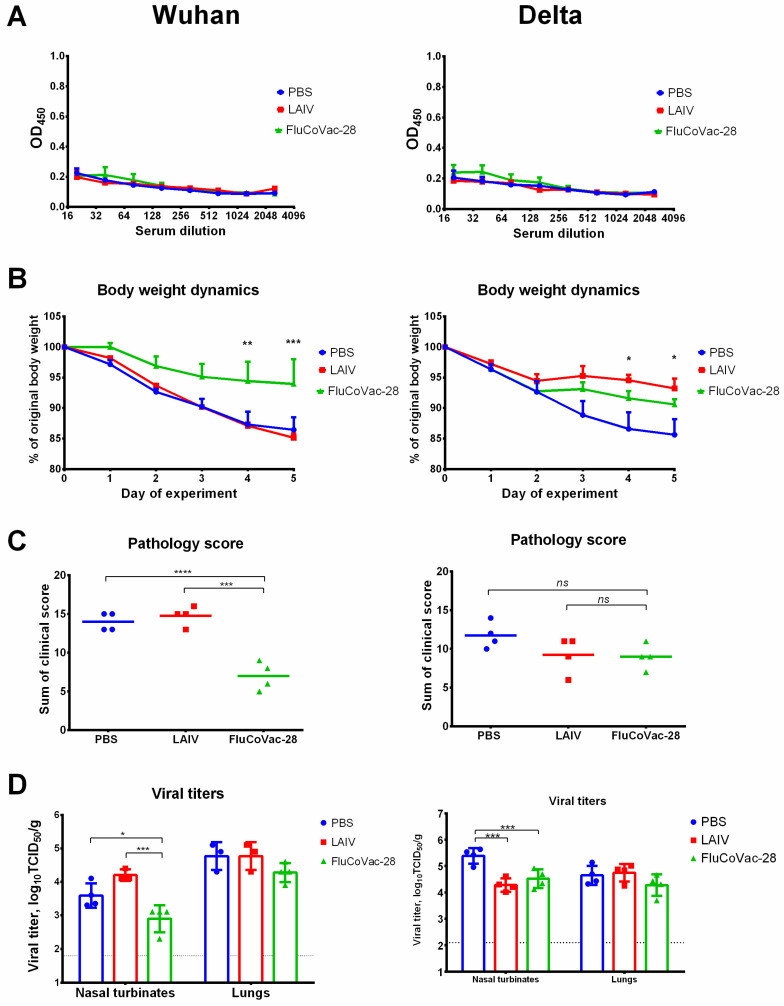
Humoral immunity and protective activity of the LAIV/SARS-CoV-2 chimeric virus against SARS-CoV-2 viruses in Syrian hamster model. Groups of animals were immunized twice with the indicated vaccine virus at a tree-week interval. Blood samples were collected 3 weeks post second dose, and then hamsters were challenged intranasally with 10^5^ TCID_50_ of SARS-CoV-2 Wuhan variant (left panel) or Delta variant (right panel). (**A**) Antibody responses to the whole SARS-CoV-2 in sera collected after two vaccine doses. (**B**) Dynamics of body weight over the challenge phase. (**C**) Sum of pathology score for 5 days post challenge. (**D**) Infectious viral titers in respiratory tissues on day 5 post challenge. Data were analyzed by two-way ANOVA with Tukey’s post-hoc multiple analyses test. *—*p* < 0.05; **—*p* < 0.01; ***—*p* < 0.001; ****—*p* < 0.0001. *ns*, not significant.

**Figure 8 vaccines-10-01142-f008:**
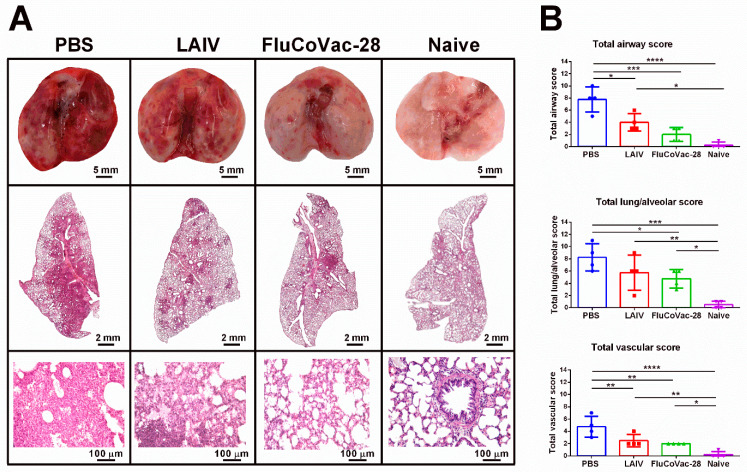
The histopathological evaluation of lung tissues of immunized Syrian hamsters on day 5 after challenge with SARS-CoV-2 Delta virus or naïve uninfected animals. (**A**) Representative micrographs of the lungs and lung sections stained with Hematoxylin & Eosin. (**B**) Semi-quantitative analyses of the changes in airway, lung/alveolar and vascular systems. Data were analyzed by one-way ANOVA with Tukey’s post-hoc multiple analyses test. *—*p* < 0.05; **—*p* < 0.01; *** —*p* < 0.001; ****—*p* < 0.0001.

**Figure 9 vaccines-10-01142-f009:**
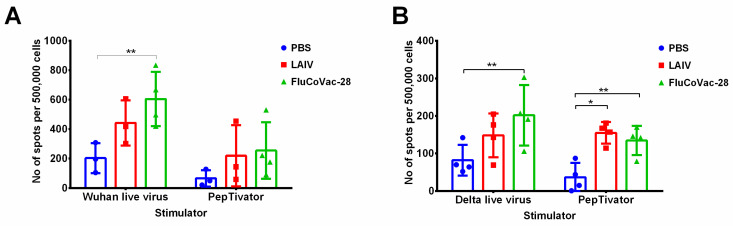
The levels of IFNγ-secreting cells in the splenocytes of immunized Syrian hamsters (*n* = 4) on day 5 after challenge with SARS-CoV-2 Wuhan (**A**) or Delta (**B**) variants. Isolated splenocytes were stimulated in vitro either with live SARS-CoV-2 or with PepTivator, followed by quantification of IFNγ-secreting cells by Hamster IFN-γ ELISpot Plus kit. Data were analyzed by two-way ANOVA with Tukey’s post-hoc multiple analyses test. *—*p* < 0.05; **—*p* < 0.01.

**Table 1 vaccines-10-01142-t001:** The replicative properties of recombinant LAIV viruses in eggs, MDCK cells and respiratory tract of HLA-A2.1 mice.

Recombinant Influenza Virus ID	SARS-CoV-2 T Cassette	Cassette Size, aa	Influenza Gene Modified	Mean Viral Titer, log_10_EID(TCID)_50_/mL			
				Eggs	MDCK Cells	Nasal Turbinates ^†^	Lungs ^†^
H7N9 LAIV	-	-	-	9.8	8.2	3.2	1.6
FluCoVac-13	Cas #1	96	NS	7.3 *	4.1 *	1.6 *	≤1.2
FluCoVac-14	Cas #2	96	NS	6.8 *	4.2 *	≤1.2 *	≤1.2
FluCoVac-15	Cas #3	122	NS	8.0 *	4.1 *	≤1.2 *	≤1.2
FluCoVac-16	Cas #4	105	NA	8.3 *	5.4 *	2.9	≤1.2
FluCoVac-17	Cas #5	122	NA	8.1 *	5.7 *	3.7	≤1.2
FluCoVac-18	Cas #6	101	NA	8.4 *	6.8 *	3.3	≤1.2
FluCoVac-27	Cas #7	161	NA	8.3 *	6.6 *	3.6	1.3
FluCoVac-28	Cas #8	174	NA	9.7	8.3	3.8	1.5
FluCoVac-29	Cas #9	122	NS	7.5 *	4.1 *	≤1.2 *	≤1.2
FluCoVac-31	Cas #10	147	NS	7.7 *	4.3 *	≤1.2 *	≤1.2
FluCoVac-32	Cas #7	161	NS	8.2 *	4.1 *	≤1.2 *	≤1.2

^†^ HLA-A2.1 mice were i.n. infected with 10^6^ EID_50_ of each influenza virus and viral titers in tissues were determined on day 3 p.i. by titration in eggs. * Significantly reduced titer (*p* < 0.05) compared to the LAIV vector virus.

## Data Availability

The data presented in this study are available on request from the corresponding author.

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
