# Peer review of "Development of a T Cell-Based COVID-19 Vaccine Using a Live Attenuated Influenza Vaccine Viral Vector"

_vaccines, 2022, doi:10.3390/vaccines10071142_

Round 1
Reviewer 1 Report
This research explains an innovative way of developing peptide based vaccine for various diseases and vaccines can be combined to target multiple diseases. There are a few point I would like to focus since they might be important for the article,
1. Line 46, it requires references.
2. I feel it would great to explain whether the mice had any types of pathological reactions to vaccines or whether the mice can be protected against SARS-Cov2 and Influenza infection since the mice have generated virus specific immune responses.
3. In figure 3, it would be great to show if there is antibody present in mice against SARS-Cov2 antigens.
4. In figure 3, it would be great to see the individual mouse antibody responses i.e. scatter plot.
5. Line 475, there will be a "with" i.e. "stimulated with sucrose-purified...".
6. It would be great if the patients who PBMCs were collected from had influenza infection or not this will show whether the vaccine could work in the presence of both infections.
7. In figure 6, it would be great to include an experiment where hamsters were challenged with both viruses i. e. influenza and SARS-Cov2 together this would show the efficacy of the vaccine is in both infections.
8. Figure 7C, is there any significant difference in viral titers when nasal turbinate and lung from individual mouse are compared since location of the virus may interfere with the viral titer.
9. Figure 9, the axis label would be "no of spots per 500,000 cells". And the author could also check whether Ova antigen could also show non-specific immune responses.
Author Response
Reviewer #1
This research explains an innovative way of developing peptide based vaccine for various diseases and vaccines can be combined to target multiple diseases. There are a few point I would like to focus since they might be important for the article,
Author’s response: we thank the reviewer for positive evaluation of our work and valuable suggestions.
- Line 46, it requires references.
Author’s response: we thank the reviewer for this note. The reference was added.
- I feel it would great to explain whether the mice had any types of pathological reactions to vaccines or whether the mice can be protected against SARS-Cov2 and Influenza infection since the mice have generated virus specific immune responses.
Author’s response: we thank the reviewer for this question. We did not assess protective effect of all recombinant viruses against influenza due to the limited number of transgenic animals available for this study. But our previous studies with similar NA- and NS-modified recombinant LAIVs demonstrated that all variants induced sterile immunity to influenza in mice. We added the following sentence to the Discussion section to address this question: “Although we did not evaluate the anti-influenza protective effect of recombinant LAIV/SARS-CoV-2 in mice in the current study because of the limited number of available animals, our previous results assume that NA- and NS-modified recombinant LAIVs can induce sterile immunity to influenza [22,23].”
As for the SARS-CoV-2 protection, mice without expression human ACE-2 are not sensitive to this virus, therefor our studies in mice were not intended to assess protection against coronavirus.
- In figure 3, it would be great to show if there is antibody present in mice against SARS-Cov2 antigens.
Author’s response: since the inserted SARS-CoV-2 cassettes were targeted for T-cell epitopes, rather than the B-cell epitopes, we did not measure antibody response to SARS-CoV-2 in mouse sera. However, the SARS-CoV-2 specific responses were assessed in immunized hamsters with the selected vaccine candidate. We included these data on Figure 7 to emphasize that the SARS-CoV-2-antibody responses were unlikely to contribute to protection.
- In figure 3, it would be great to see the individual mouse antibody responses i.e. scatter plot.
Author’s response: we thank the reviewer for this suggestion. The figure was changed to show individual data for each animal. In addition, figures 4, 6, 7, 9 were also changed to show the individual values.
- Line 475, there will be a "with" i.e. "stimulated with sucrose-purified...".
Author’s response: we thank the reviewer for noting this typo. It was corrected accordingly.
- It would be great if the patients who PBMCs were collected from had influenza infection or not this will show whether the vaccine could work in the presence of both infections.
Author’s response: all volunteers in the study had a history of exposure to influenza antigen and some of them had been immunized with inactivated vaccines. LAIV immunization was an exclusion criterion for participation in the study. In this study, we compared the response to the LAIV antigen alone (i.e., the viral vector) with the response to recombinant LAIV viruses containing SARS-CoV-2 insertions, and we considered the differences in responses to the two stimulating agents as a response to the SARS-CoV-2 inserted epitopes. In our study, the LAIV vector was based on the H7N9 influenza virus, which did not circulate in humans, so memory T-cell responses were not expected to be significant among our participants. In addition, the LAIV was based on Leningrad/17 H2N2 virus, and H2N2 viruses have not circulated since 1968. All of our participants were born after 1968, so T-cell responses to internal proteins were also expected to be low. We included this paragraph in the Discussion section.
- In figure 6, it would be great to include an experiment where hamsters were challenged with both viruses i. e. influenza and SARS-Cov2 together this would show the efficacy of the vaccine is in both infections.
Author’s response: we thank the reviewer for this suggestion. Indeed, the effect of a bivalent vaccine on protection against co-infection needs to be studied in detail. However, in this first report on the development of such a vaccine, it was important to demonstrate the effect against each infection separately to ensure that the vaccine could be effective if the vaccinated person were to become infected at different points in time, which is more likely than infection with two viruses simultaneously. We added this statement to the Conclusions section.
- Figure 7C, is there any significant difference in viral titers when nasal turbinate and lung from individual mouse are compared since location of the virus may interfere with the viral titer.
Author’s response: we did not observe significant differences between viral titers in nasal turbinates and lungs among challenged hamsters. We added individual titer values to the Figure 7 to demonstrate that the titers were comparable in animals within each test group.
- Figure 9, the axis label would be "no of spots per 500,000 cells". And the author could also check whether Ova antigen could also show non-specific immune responses.
Author’s response: we thank the referee for this note. The axis labels were corrected accordingly on Figures 9 and 4.
Reviewer 2 Report
The Isakova-Sivak et al. study describes the development of a T cell-based COVID-19 vaccine using a live-attenuated influenza vaccine viral vector. The authors generated polyepitope cassettes based on the SARS CoV-2 S, N, and M protein fragments enriched with selected human T cell epitopes. The 10 cassettes were cloned into neuraminidase (NA) or non-structural (NS) genes of the H7N9 LAIV strain-based virus individually. The mouse study tested all the 11 vaccines generated and resulted in no detectable SARS CoV-2-specific CD8+T cell responses which the authors attribute to the immunodominance of the influenza-specific epitopes. Based on the in-vitro studies utilizing PBMCs from COVID-19 convalescent subjects, they picked a single vaccine FluCoVac-28 to test in hamsters. The FluCoVac-28 and control LAIV vaccinated hamsters were divided into 3 groups and challenged with either SH/PR8influenza virus or SARS CoV-2 Wuhan lineage or SARS- Cov-2 Delta lineage. The. Flu challenge resulted in significantly reduced viral titers in the lungs and nasal turbinates of the FluCoVac-28 and LAIV vaccinated hamsters. The Wuhan lineage challenge resulted in significantly reduced weight loss, lung pathology score, and infectious viral titers in the nasal turbinates but no significant reduction in the lung infectious viral titers was observed in the FluCoVac-28 group. The Delta challenge resulted in an odd result wherein the LAIV control and the FluCoVac-28 group both showed reduced weight loss in comparison to the control PBS, with no difference in the Lung pathology score and the lung viral infectious titers, but significantly reduced viral infectious titers in the nasal turbinates. Despite the presence of the virus in the lungs, the FluCoVac-28 showed significantly reduced inflammation in the lungs in the Wuhan and Delta lineage challenge studies.
This study has some interesting data, but the hamster challenge study results are rather confusing and inconclusive.
Major concerns:
1) Why did the authors only pick FluCoVac-28 and not test #29, 31, and 17 in addition, as a combined vaccine? Have you considered using another vector that can carry multiple polyepitopes?
2) Which strain of infection did the COVID-19 convalescent subjects have (from whom PBMCs were collected)?
3) It is well known that a SARS CoV-2 vaccine needs to induce antibodies in order to protect against disease. T cell responses might augment protection- why did the authors not have an arm of an antibody-based vaccine in addition to their T cell-based vaccine to verify this?
4) Considering the LAIV vaccinated animals were protected against the Delta strain challenge in terms of weight loss, and pathology score, it is unclear if the protection seen in FluCoVac-28 vaccinated animals is due to unspecific T cell activation (which may fade with time) or if it is specific. How can such a vaccine protect long-term?
5) Could the influenza immunodominance response seen in HLA-A*0.2:1 mice also be seen in humans?
6) Did the authors analyze the hamster lung chemokine and cytokine responses (besides IFN-gamma) post-challenge?
7) Since the splenocytes of LAIV vaccinated hamsters secrete similar/high levels of IFN-gamma post-challenge similar to the FluCoVac-28 group, it is difficult to conclude that the FluVoVac vaccine has a significant advantage in reducing immune-mediated inflammation and pathology. Can the authors' comment.
Minor Concerns
1) The vaccination and challenge information is not listed in the results section and the reader has to keep scanning the methods section for that information.
2) Figure legends should have more detailed information eg: challenge, titer, day of the challenge, dose of vaccine, etc.
3) The authors should provide histology images for all the animals in the study in the supplementary section.
4) Why didn't the authors have a Mock challenged group for normal hamster lungs histology as a comparator.
Author Response
Reviewer #2.
The Isakova-Sivak et al. study describes the development of a T cell-based COVID-19 vaccine using a live-attenuated influenza vaccine viral vector. The authors generated polyepitope cassettes based on the SARS CoV-2 S, N, and M protein fragments enriched with selected human T cell epitopes. The 10 cassettes were cloned into neuraminidase (NA) or non-structural (NS) genes of the H7N9 LAIV strain-based virus individually. The mouse study tested all the 11 vaccines generated and resulted in no detectable SARS CoV-2-specific CD8+T cell responses which the authors attribute to the immunodominance of the influenza-specific epitopes. Based on the in-vitro studies utilizing PBMCs from COVID-19 convalescent subjects, they picked a single vaccine FluCoVac-28 to test in hamsters. The FluCoVac-28 and control LAIV vaccinated hamsters were divided into 3 groups and challenged with either SH/PR8influenza virus or SARS CoV-2 Wuhan lineage or SARS- Cov-2 Delta lineage. The. Flu challenge resulted in significantly reduced viral titers in the lungs and nasal turbinates of the FluCoVac-28 and LAIV vaccinated hamsters. The Wuhan lineage challenge resulted in significantly reduced weight loss, lung pathology score, and infectious viral titers in the nasal turbinates but no significant reduction in the lung infectious viral titers was observed in the FluCoVac-28 group. The Delta challenge resulted in an odd result wherein the LAIV control and the FluCoVac-28 group both showed reduced weight loss in comparison to the control PBS, with no difference in the Lung pathology score and the lung viral infectious titers, but significantly reduced viral infectious titers in the nasal turbinates. Despite the presence of the virus in the lungs, the FluCoVac-28 showed significantly reduced inflammation in the lungs in the Wuhan and Delta lineage challenge studies.
This study has some interesting data, but the hamster challenge study results are rather confusing and inconclusive.
Author’s response: we thank the reviewer for her critical appraisal of our manuscript and all valuable suggestions that helped us improve its quality.
Major concerns:
1) Why did the authors only pick FluCoVac-28 and not test #29, 31, and 17 in addition, as a combined vaccine? Have you considered using another vector that can carry multiple polyepitopes?
Author’s response: we selected this particular variant out of all vaccine candidates tested because of several preferred characteristics, such as its ability to actively replicate in mammalian cells (a very important property for a vaccine that can be produced on MDCK cells in the future), its high reproductive activity in the upper respiratory tract of animals, and because this variant encodes the cassette carrying the highest number of SARS-CoV-2 T-cell epitopes. We agree with the reviewer that other variants were also promising based on the in vitro tests on PBMCs, but the limited capacity of the animal BSL-3 facility allowed testing only one candidate, along with the vector control, in the challenge experiments.
We did not consider other viral vectors (e.g., PIV-3 or AdV) to develop a T-cell-based SARS-CoV-2 vaccine because we are trying to develop a bivalent vaccine against SARS-CoV-2 and influenza, and the influenza virus-based vector mediates protection against the latter infection. In this pilot study, we tested only variants carrying a single polyepitope cassette, but it is possible to create recombinant viruses carrying several cassettes in different viral genes, which can significantly increase the protective potential against SARS-CoV-2. We added this statement in the Conclusions section.
2) Which strain of infection did the COVID-19 convalescent subjects have (from whom PBMCs were collected)?
Author’s response: the PBMCs were collected from COVID-19 patients with disease onset from March 2020 till August 2020, which means that the infection was caused by the Wuhan D614G variant which circulated in Saint Petersburg at that time. We added the following sentence to the Materials and Methods section: “During this period, the Wuhan D614G SARS-CoV-2 variant dominated circulation in Russia, suggesting that study participants were infected with this strain”.
3) It is well known that a SARS CoV-2 vaccine needs to induce antibodies in order to protect against disease. T cell responses might augment protection- why did the authors not have an arm of an antibody-based vaccine in addition to their T cell-based vaccine to verify this?
Author’s response: we agree with the reviewer that antibodies are essential in inducing sterilizing immunity, whereas T cells can facilitate the recovery. However, antibody-induced immunity is very sensitive to the mutations in viral antigens and therefore the virus can easily escape from the action of specific antibodies. In contrast, T cells are induced to more conserved epitopes and they can recognize antigenically diverse viruses and therefore augment protection against various variants of concern. As stated in Altmann and Boyton (doi: 10.1126/sciimmunol.abd6160), “A general observation from the patient cohorts is that most infected people make an antibody and a T cell response, the magnitude of the two often being correlated. However, it is also the case that in SARS-CoV-2 infection, measures of T cell and B cell recognition can become uncoupled, either because mild infection has triggered T cell immunity without detectable antibody or because the antibody response has been transient and already waned at a time when T cell memory is still robust. It is clear from these datasets that some people who lack an antibody response (and indeed may never have been formally defined as PCR+) show strong, specific T cell immunity”. These data justify that the T-cell responses are generally longer-lived than antibody responses, and given that the T cells induced by vaccination have a potential to recognize similar pathogens, such T cell-based vaccines can facilitate viral clearance even in the absence of virus-specific antibody. We added the following sentence to the Discussion section to emphasize the prospect of T cell-based vaccine development: “Furthermore, there is evidence that soon after infection with SARS-CoV-2, especially the mild form, people do not have detectable virus-specific antibodies, although T-cell im-munity is quite robust, either because no antibodies have been produced or they were very short-lived, whereas T-cell memory lasts much longer [53]”
Nevertheless, our team has also generated a panel of LAIV-based bivalent vaccine candidates expressing Spike RBD fragment of SARS-CoV-2 to induce antibody-mediated protection, and a separate manuscript describing these constructs is currently under preparation. An ultimate goal of the designing an efficacious bivalent vaccine is the induction of robust antibody and T-cell responses to SARS-CoV-2 afforded by the delivery of the selected B- and T-cell cassettes to target cells using LAIV viral vector. We added the corresponding statement to the Conclusions section.
4) Considering the LAIV vaccinated animals were protected against the Delta strain challenge in terms of weight loss, and pathology score, it is unclear if the protection seen in FluCoVac-28 vaccinated animals is due to unspecific T cell activation (which may fade with time) or if it is specific. How can such a vaccine protect long-term?
Author’s response: we thank the reviewer for this important question. We assume that there is still a specific response - according to ELISPOT data, the number of INFγ-producing cells was higher in the bivalent vaccine group than in the LAIV group when stimulated with live coronavirus. However, some level of non-specific activation could have contributed to the protection against SARS-CoV-2 seen in the LAIV group. The different level of non-specific protection of LAIV against Wuhan and Delta virus observed in the hamster challenge experiments can be explained by different pathogenicity of the two SARS-CoV-2 variants for these animals, as we noted the milder course of the disease after infection with Delta strain, compared to the Wuhan virus, and the non-specific activation of T cells could be more effective in case of milder infection.
Similar results of non-specific protection were observed by other researchers who developed LAIV-based SARS-CoV-2 vaccine (RBD expressed within the NS1 open reading frame) (doi: 10.1016/j.scib.2022.05.018). This group also had nonspecific protection by the empty vector in the challenge experiment, and there were no antibodies detectable by routine ELISA, although their vaccine is not polyepitope and could have stimulated antibody production. This group has shown that a local immune response plays a role, and they demonstrated that vaccination with this type of vaccine provides long-term protection. We also plan to conduct similar experiments in the future.
We added these statements to the Discussion section.
5) Could the influenza immunodominance response seen in HLA-A*0.2:1 mice also be seen in humans?
Author’s response: the phenomenon of immunodominance is particularly pronounced in inbred animals, precisely due to their identical haplotype. In nature, where animals are not inbred, immunodominance continues to exist but does not dramatically affect the results of experiments. Since humans are very heterogeneous by their HLA genotypes, some epitopes will be more immunogenic, some less immunogenic, but using a large number of SARS-CoV-2 epitopes (and the selected candidate contains the maximum number of all tested cassettes) will probably result in a significant percentage of the population having something immunogenic for that particular individual. We also added the following sentence to the Discussion section to emphasize that the magnitude of responses even to the same epitopes can be different in transgenic mice and humans: “For example, a side-by-side screening of T-cell epitopes in HLA-A2.1 transgenic mice and HLA-A2.1-positive subjects who received a vaccine against vaccinia virus found that only 46% epitopes were detected in both systems, and the responses in transgenic mice against epitopes originally identified in mice were much stronger than for epitopes originally detected in humans, and vice versa [70].”
6) Did the authors analyze the hamster lung chemokine and cytokine responses (besides IFN-gamma) post-challenge?
Author’s response: we thank the reviewer for the interesting point. Unfortunately, due to the limited availability of hamster reagents we were unable to test lung chemokine and memory T-cell responses by flow cytometry. We plan to perform deeper evaluation of the safety and immunogenicity of our LAIV/SARS-CoV-2 candidates in relevant pre-clinical models in the near future.
7) Since the splenocytes of LAIV vaccinated hamsters secrete similar/high levels of IFN-gamma post-challenge similar to the FluCoVac-28 group, it is difficult to conclude that the FluVoVac vaccine has a significant advantage in reducing immune-mediated inflammation and pathology. Can the authors' comment.
Author’s response: we agree with the reviewer that LAIVs induced cross-reactive T-cell responses to SARS-CoV-2, which has contributed to protection against Delta variant, a virus that was less pathogenic in the hamster model than the Wuhan strain. However, in the Wuhan challenge experiment, the LAIV itself was unable to protect animals against viral replication and lung pathology, although IFNγ responses were also noted in this group. The major issue of defining significant differences between study groups come from the low number of animals in each test group, but the overall data from the hamster challenge study favors a bivalent vaccine.
Minor Concerns
1) The vaccination and challenge information is not listed in the results section and the reader has to keep scanning the methods section for that information.
Author’s response: we added the missing information to the Results section (paragraphs 3.2, 3.5).
2) Figure legends should have more detailed information eg: challenge, titer, day of the challenge, dose of vaccine, etc.
Author’s response: we included more detailed information to the figure legends.
3) The authors should provide histology images for all the animals in the study in the supplementary section.
Author’s response: we added the images of lung sections stained with H&E in the Supplementary section (Figures S14 and S15).
4) Why didn't the authors have a Mock challenged group for normal hamster lungs histology as a comparator.
Author’s response: we had the control group of uninfected hamsters to identify the viral-specific tissue damages in the lungs, but did not include this group in original version of the manuscript since we were focused on the test groups that were challenged with SARS-CoV-2. We added this group to the Figure 8 as requested by the reviewer. It should be noted that the results of the statistical analysis changed slightly because of this additional group.
Reviewer 3 Report
Review comments on vaccines-1775091
Ms. Ref. No. vaccines-1775091
Title: Development of a T cell-based COVID-19 vaccine using a live 2 attenuated influenza vaccine viral vector
Authors: Irina Isakova-Sivak, Ekaterina Stepanova, Victoria Matyushenko et al.
Major comments:
Current COVID-19 pandemic has caused an unprecedented burden to all people in the world. SARS-CoV-2 virus continues to circulate and antigenically evolve enabling multiple re-infections. This study was conducted to address the issue of the virus antigenic variability and to develop T cell-based vaccines directed to more conserved viral epitopes from internal viral proteins. The authors used live attenuated influenza vaccine (LAIV) virus vector to generate recombinant influenza viruses expressing various T-cell epitopes of SARS-CoV-2 dfrom either NA or NS1 genes (NA, neuraminidase; NS1, non-structural protein), by introducing P2A self-cleavage site. They found that intranasal immunization of HLA-A2.1 transgenic mice with these recombinant viruses didn’t result in significant SARS-CoV-2-specific T-cell responses, due to the immunodominance of NP366 epitope of the influenza virus. However, stimulation of peripheral blood mononuclear cells (PBMCs) of COVID-19 convalescents with various forms of recombinant viruses and LAIV vector alone demonstrated the activation of memory T cells in samples stimulated with LAIV/SARS-CoV-2, but not LAIV alone. Hamsters immunized with a selected LAIV/SARS-CoV-2 virus (FluCoVac-28) were found to be protected against challenge with influenza virus and a high dose of SARS-CoV-2 of Wuhan and Delta lineages, based on several pieces of evidence; such as reduced weight loss, milder clinical symptoms and less pronounced histopathological sighs of SARS-CoV-2 infection in the lungs, in comparison with those of LAIV- and mock-immunized animals. Based on these results, the authors concluded that LAIV is a promising platform for the development of a bivalent vaccine against both influenza and SARS-CoV-2.
The experiments were well-planned and conducted and the results were clear. They also provided many supporting results in the supporting materials. The conclusions, which the authors made based on these results, are scientifically sound. Accordingly, this manuscript is acceptable for publication in “Vaccines” after appropriate revisions against following minor comments.
Minor comments:
(1) Although most of abbreviated words are spelled out in their first appearance, some words are used without definition. They should be spelled out in their first appearance in the manuscript. They include CTL (cytotoxic T lymphocyte?), MHC (major histocompatibility complex), HA (hemagglutination), HAI assay (hemagglutination inhibition assay?), PSO (?), PMA (phorbol myristate acetate), MOI(?).
(2) Although the authors stated that “only one out of 11 chimeric viruses (FluCoVac-28) had identical titer to the LAIV vector” (page 10, line 417), it was actually very similar (9.7 vs 9.8) (Table 1).
(3) In Figure 2, generation of recombinant LAIV viruses expressing SARS-CoV-2 T cell cassettes is shown in each panel A and B, respectively. However, in the left part, in circles, the genome composition of the recombinant LAIV viruses is shown but without explanation in Figure legend.
Author Response
Reviewer #3.
Review comments on vaccines-1775091
Ms. Ref. No. vaccines-1775091
Title: Development of a T cell-based COVID-19 vaccine using a live 2 attenuated influenza vaccine viral vector
Authors: Irina Isakova-Sivak, Ekaterina Stepanova, Victoria Matyushenko et al.
Major comments:
Current COVID-19 pandemic has caused an unprecedented burden to all people in the world. SARS-CoV-2 virus continues to circulate and antigenically evolve enabling multiple re-infections. This study was conducted to address the issue of the virus antigenic variability and to develop T cell-based vaccines directed to more conserved viral epitopes from internal viral proteins. The authors used live attenuated influenza vaccine (LAIV) virus vector to generate recombinant influenza viruses expressing various T-cell epitopes of SARS-CoV-2 dfrom either NA or NS1 genes (NA, neuraminidase; NS1, non-structural protein), by introducing P2A self-cleavage site. They found that intranasal immunization of HLA-A2.1 transgenic mice with these recombinant viruses didn’t result in significant SARS-CoV-2-specific T-cell responses, due to the immunodominance of NP366 epitope of the influenza virus. However, stimulation of peripheral blood mononuclear cells (PBMCs) of COVID-19 convalescents with various forms of recombinant viruses and LAIV vector alone demonstrated the activation of memory T cells in samples stimulated with LAIV/SARS-CoV-2, but not LAIV alone. Hamsters immunized with a selected LAIV/SARS-CoV-2 virus (FluCoVac-28) were found to be protected against challenge with influenza virus and a high dose of SARS-CoV-2 of Wuhan and Delta lineages, based on several pieces of evidence; such as reduced weight loss, milder clinical symptoms and less pronounced histopathological sighs of SARS-CoV-2 infection in the lungs, in comparison with those of LAIV- and mock-immunized animals. Based on these results, the authors concluded that LAIV is a promising platform for the development of a bivalent vaccine against both influenza and SARS-CoV-2.
The experiments were well-planned and conducted and the results were clear. They also provided many supporting results in the supporting materials. The conclusions, which the authors made based on these results, are scientifically sound. Accordingly, this manuscript is acceptable for publication in “Vaccines” after appropriate revisions against following minor comments.
Author’s response: we thank the reviewer for positive evaluation of our work.
Minor comments:
(1) Although most of abbreviated words are spelled out in their first appearance, some words are used without definition. They should be spelled out in their first appearance in the manuscript. They include CTL (cytotoxic T lymphocyte?), MHC (major histocompatibility complex), HA (hemagglutination), HAI assay (hemagglutination inhibition assay?), PSO (?), PMA (phorbol myristate acetate), MOI(?).
Author’s response: we thank the reviewer for this critique. All abbreviations were spelled out in their first appearance.
(2) Although the authors stated that “only one out of 11 chimeric viruses (FluCoVac-28) had identical titer to the LAIV vector” (page 10, line 417), it was actually very similar (9.7 vs 9.8) (Table 1).
Author’s response: we thank the reviewer for this note. The sentence was corrected accordingly.
(3) In Figure 2, generation of recombinant LAIV viruses expressing SARS-CoV-2 T cell cassettes is shown in each panel A and B, respectively. However, in the left part, in circles, the genome composition of the recombinant LAIV viruses is shown but without explanation in Figure legend.
Author’s response: we thank the reviewer for this comment. The figure legend was corrected to make the figure clearer to the readers.
Round 2
Reviewer 2 Report
The authors have addressed my concerns appropriately and incorporated the changes suggested.